# Coral anthozoan-specific opsins employ a novel chloride counterion for spectral tuning

Yusuke Sakai[1], Saumik Sen[2,3], Tomohiro Sugihara[1], Yukiya Kakeyama[1], Makoto Iwasaki[1], Gebhard FX Schertler[4], Xavier Deupi[2,3,4]*, Mitsumasa Koyanagi[1,5]*, Akihisa Terakita[1,5]*

[1]Department of Biology, Graduate School of Science, Osaka Metropolitan University, Sumiyoshi ku, Osaka, Japan; [2]Swiss Institute of Bioinformatics (SIB), Lausanne, Switzerland; [3]Condensed Matter Theory Group, Laboratory for Theoretical and Computational Physics, PSI Center for Scientific Computing, Theory, and Data, Villigen PSI, Switzerland; [4]Laboratory of Biomolecular Research, PSI Center for Life Sciences, Villigen PSI, Switzerland; [5]The OMU Advanced Research Institute for Natural Science and Technology, Osaka Metropolitan University, Sumiyoshi-ku, Osaka, Japan

*For correspondence:
xavier.deupi@psi.ch (XD);
koyanagi@omu.ac.jp (MK);
terakita@omu.ac.jp (AT)

Competing interest: The authors declare that no competing interests exist.

## eLife Assessment

The authors provide **compelling** evidence that a chloride ion stabilizes the protonated Schiff base chromophore linkage in the animal rhodopsin Antho2a. This **important** finding is novel and of major interest to a broad audience, including optogenetics researchers, protein engineers, spectroscopists, and environmental biologists. The study combines state-of-the-art research methods, such as spectroscopic and mutational analyses, which are complemented by QM/MM calculations, and was further improved based on the comments from the reviewers.

**Abstract** Animal opsins are G protein-coupled receptors that have evolved to sense light by covalently binding a retinal chromophore via a protonated (positively charged) Schiff base. A negatively charged amino acid in the opsin, acting as a counterion, stabilizes the proton on the Schiff base, which is essential for sensitivity to visible light. In this study, we investigate the spectroscopic properties of a unique class of opsins from a reef-building coral belonging to the anthozoan-specific opsin II group (ASO-II opsins), which intriguingly lack a counterion residue at any of established sites. Our findings reveal that, unlike other known animal opsins, the protonated state of the Schiff base in visible light-sensitive ASO-II opsins is highly dependent on exogenously supplied chloride ions ($Cl^-$). By using structural modeling and quantum mechanics/molecular mechanics (QM/MM) calculations to interpret spectroscopy data, we conclude that, in the dark state, ASO-II opsins employ environmental $Cl^-$ as their native counterion, while a nearby polar residue, Glu292 in its protonated neutral form, facilitates $Cl^-$ binding. In contrast, Glu292 plays a crucial role in maintaining the protonation state of the Schiff base in the light-activated protein, serving as the counterion in the photoproduct. Furthermore, Glu292 is involved in G protein activation of the ASO-II opsin, suggesting that this novel counterion system coordinates multiple functional properties.

## Introduction

Animals sense light by using opsins, photosensitive proteins belonging to the large family of G protein-coupled receptors (GPCRs). These proteins have a seven-transmembrane helix structure and bind to a retinal chromophore to form a light-sensitive pigment. Opsins are present in the genomes of all eumetazoans (i.e. all animal lineages except sponges), and based on their phylogenetic relationships, they can be classified into eight groups with distinctive properties: vertebrate visual/nonvisual opsins, opn3/TMT opsins, invertebrate Go-coupled opsins, cnidarian Gs-coupled opsins (cnidopsins), neuropsins (opn5), Gq-coupled visual pigments/melanopsins (opn4), peropsins, and retinochrome/RGR (*Koyanagi and Terakita, 2014*). Such diversity possibly underlies the diversification of light-dependent physiologies in animals. Furthermore, this diversity also provides a range of potential optogenetic tools to manipulate intracellular G protein-mediated signaling (*Koyanagi and Terakita, 2014*).

Reef-building corals and sea anemones belong to the subphylum Anthozoa, which together with the subphylum Medusozoa constitute the phylum Cnidaria. Cnidarian animals possess multiple opsins categorized as part of the cnidarian Gs-coupled opsin group (cnidopsins), which regulate light-dependent processes (*Koyanagi et al., 2008*). For example, a member of this group is expressed in the ciliary-type visual cells of the box jellyfish lens eyes (*Koyanagi et al., 2008*; *Kozmik et al., 2008*). Beyond these Gs-coupled cnidopsins, anthozoan animals have opsins that are phylogenetically distinct from the other known eight groups and are found exclusively in anthozoans (*Feuda et al., 2012*; *Suga et al., 2008*). These anthozoan-specific opsins (ASO) can be further classified into two groups, ASO-I and ASO-II (*Gornik et al., 2021*; *Hering and Mayer, 2014*; *Mason et al., 2023*; *Picciani et al., 2018*; *Ramirez et al., 2016*). *Gornik et al., 2021*, proposed that both ASO-I and ASO-II were present in the last common ancestor of Anthozoa and Medusozoa but were lost secondarily in the Medusozoa lineage (*Gornik et al., 2021*). While it has been reported that both ASO-I and ASO-II are expressed in multiple tissues of sea anemones (*Gornik et al., 2021*; *Suga et al., 2008*) and corals (*Levy et al., 2021*), there is still a limited understanding of their molecular characteristics and physiological functions.

The members of the ASO-II group are not only phylogenetically unique but also display interesting features in their amino acid sequences. For instance, several of these opsins lack an amino acid residue conserved among typical opsins that is crucial for absorption of visible light (*Gornik et al., 2021*; *Mason et al., 2023*). While free retinal in solution has its absorption maximum ($\lambda_{max}$) in the ultraviolet (UV), this shifts to visible light when retinal is bound to a lysine residue in the transmembrane bundle of the opsin (usually at Lys296, numbering according to the bovine rhodopsin sequence) through a protonated Schiff base to form the pigment. Such a protonated Schiff base is necessary to achieve sensitivity to visible light in opsin-based pigments (*Pitt et al., 1955*). However, the proton on the positively charged Schiff base is energetically unstable in the hydrophobic transmembrane environment. To stabilize this proton, a negatively charged residue, glutamic or aspartic acid, is situated near the Schiff base to act as a counterion. This counterion is essential for opsin-based pigments to absorb visible light, and the residues serving as the counterion are highly conserved across opsins (*Nathans, 1990*; *Terakita, 2005*; *Terakita et al., 2012*). To date, three experimentally confirmed sites for the counterion have been identified in animal opsins: 94 in helix 2 (*Gerrard et al., 2018*), 113 in helix 3 (*Nathans, 1990*; *Sakai et al., 2022*; *Sakmar et al., 1989*; *Zhukovsky and Oprian, 1989*), and 181 in extracellular loop 2 (*Nagata et al., 2019*; *Terakita et al., 2004*; *Terakita et al., 2000*). Remarkably, some opsins belonging to the ASO-II group lack glutamic or aspartic acid at any of these established counterion positions (*Gornik et al., 2021*; *Mason et al., 2023*). This absence raises the question of whether these opsins can absorb visible light, and if so, by what mechanism.

In this study, we investigate the spectroscopic properties of opsins in the ASO-II group isolated from the reef-building coral *Acropora tenuis*. Absorption spectra reveal that this group includes opsins sensitive to both UV and visible light. We then focus on a particular visible light-sensitive opsin within the ASO-II group (Antho2a) by spectroscopically analyzing the protonated and deprotonated states of the Schiff base in the wild-type and in single-point mutants. By interpreting the spectroscopy data in the light of hybrid quantum mechanics/molecular mechanics (QM/MM) simulations, we demonstrate that a chloride anion ($Cl^-$) serves as a counterion to the retinylidene Schiff base in animal opsins, specifically in visible light-sensitive opsins of the ASO-II group.

## Results

### Identification of *A. tenuis* opsins

We identified 17 opsins from the *A. tenuis* genome and transcriptome datasets by homology search, which included eight opsins in Gs-coupled cnidopsin group, one opsin in the ASO-I group, and eight opsins in the ASO-II group (*Figure 1A*; *Figure 1—figure supplement 1*). Full-length cDNAs of seven out of the eight opsins in the ASO-II group were isolated and cloned from adult or larval tissues of the coral (highlighted by bold letters in *Figure 1A*). We failed to amplify one opsin in the ASO-II group (gene model ID in the OIST Marine Genomics Unit Genome Project; *Shinzato et al., 2021*: aten_s0263.g14) by RT-PCR possibly because of its little mRNA expression level. Amino acid sequence alignment shows that all the seven *A. tenuis* opsins in the ASO-II group lack a glutamic or aspartic acid at the established counterion positions 94, 113, or 181 (*Figure 1B*; *Figure 1—figure supplement 2*). These opsins also have no E(D)RY motif at the cytoplasmic end of helix 3 (*Figure 1—figure supplement 2*), which is conserved throughout most class A GPCRs (*Hofmann et al., 2009*).

### Absorption spectra of *A. tenuis* opsins in the ASO-II group

We expressed seven members of the ASO-II group in COS-1 cells and purified their recombinant pigments in detergent-solubilized conditions. We successfully obtained the absorption spectra of three (Antho2a, Antho2c, and Antho2e) out of the seven members, which showed that Antho2a and Antho2c are visible light-sensitive opsins with $\lambda_{max}$ at 503 nm and 450 nm, respectively, whereas Antho2e is a UV-sensitive opsin with $\lambda_{max}$ at ~360 nm (*Figure 1C*). We have previously reported that one opsin in the ASO-II group, acropsin 4 of the coral *Acropora millepora*, induces a light-dependent elevation of intracellular $Ca^{2+}$ levels (*Mason et al., 2023*). Here, we showed that Antho2a, Antho2c, and Antho2e evoked a similar light-dependent increase of $Ca^{2+}$ levels in HEK293S cells (*Figure 1D*).

### Search for the counterion in *A. tenuis* opsins of the ASO-II group

Antho2a and Antho2c form visible light-sensitive pigments in the dark (*Figure 1C*) despite the lack of a negatively charged counterion at any of the established positions (*Figure 1B*; *Figure 1—figure supplement 2*). To investigate how the protonated Schiff base is stabilized in these opsins, we studied in more detail Antho2a ($\lambda_{max}$ = 503 nm), as it could be expressed well in cultured cells and was stable in detergent-solubilized conditions (*Figure 1C*).

### Contribution of Glu292 to the absorption spectra of the dark state and photoproduct of Antho2a

First, we searched for potential counterions at positions different from known established amino acid sites (91, 113, and 181) in the Antho2a sequence. Using the crystal structure of bovine rhodopsin (PDB ID: 1U19) as a template, we identified glutamic or aspartic acids located within 5 Å of the Schiff base in Antho2a and other members in the ASO-II group. Notably, all *A. tenuis* opsins in this group contain a conserved glutamic/aspartic acid at position 292 (*Figure 1B*; *Figure 1—figure supplement 2*), positioned just one helix turn away from the retinal-binding residue Lys296. To determine whether Glu292 could function as the counterion in Antho2a, we mutated Glu292 to alanine and measured the absorption spectra. The absorption spectrum of the E292A mutant in the dark was nearly identical to that of wild type (*Figure 2A and B*, curve 1), exhibiting a clear absorbance in the visible light region with only a slightly red-shifted $\lambda_{max}$ (505 nm) at 140 mM NaCl and pH 6.5. This shows that a negative charge other than Glu292 may serve as a counterion in the dark state of wild-type Antho2a.

We next investigated the spectroscopic properties of the photoproduct of wild-type Antho2a and the E292A mutant. Upon irradiation of wild-type Antho2a with orange light, the $\lambda_{max}$ shifted from 503 nm in the dark to 476 nm in the photoproduct (*Figure 2A*, curve 2; *Figure 2—figure supplement 1A*, curves 2 and 3). This shift is due to the photoisomerization of the 11-*cis* retinal chromophore to its all-*trans* form, converting almost 100% of the dark state to the photoproduct (*Figure 2—figure supplement 1B*). The photoproduct remained stable for at least 5 min (*Figure 2—figure supplement 1A*, curves 2 and 3) but did not revert to the original dark state upon subsequent irradiation (*Figure 2—figure supplement 1A and C*). Instead, it underwent gradual decay accompanied by retinal release over time (*Figure 2—figure supplement 1D–G*). These findings indicate that purified Antho2a is neither strictly bleach resistant nor bistable (see also *Figure 2—figure supplement 1* legend). We

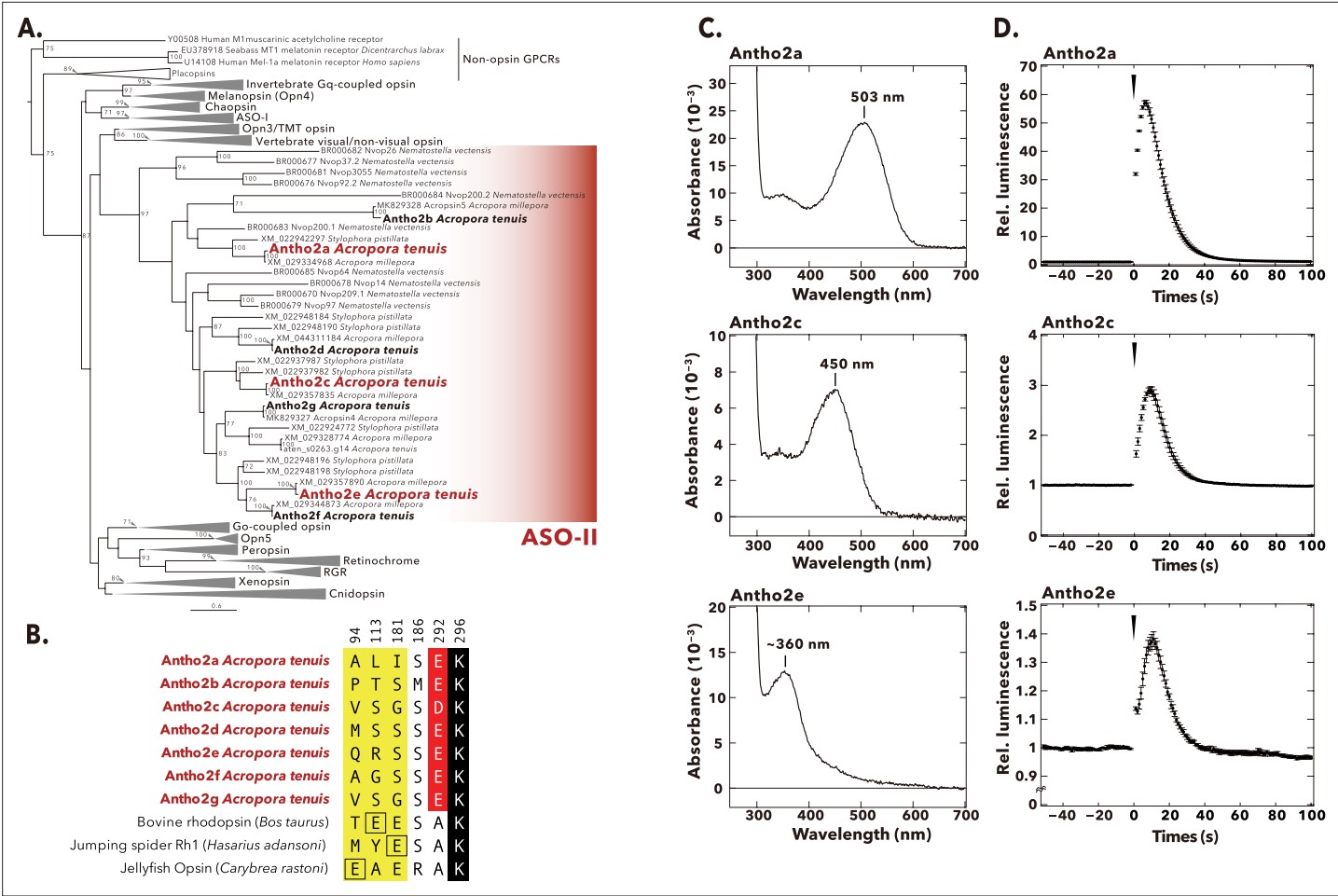

**Figure 1.** Phylogenetic tree, selected amino acid residues, absorption spectra, and light-induced Ca²⁺ responses of *A. tenuis* opsins belonging to the anthozoan-specific opsin II (ASO-II) group. (**A**) Maximum-likelihood (ML) tree of animal opsins including *A. tenuis* opsins in the ASO-II group. Seven opsins in the ASO-II group that were identified and cloned from *A. tenuis* in this study are shown in bold, and the three members for which we obtained absorption spectra are highlighted in red. Numbers at the nodes represent support values of each ML branch estimated by 1000 bootstrap samplings (≥70% are indicated). Scale bar = 0.6 substitutions per site. All branches and support values are provided in *Figure 1—figure supplement 1*. (**B**) Selected residues near the Schiff base in opsins of the ASO-II group and other animal opsins. Animal opsins typically have an acidic residue acting as counterion at one of three established sites (yellow): E94 (e.g. jellyfish opsin), E113 (e.g. bovine rhodopsin), or E181 (e.g. jumping spider Rh1). Remarkably, opsins in the ASO-II group lack an acidic residue at any of these positions but instead feature an acidic residue at position 292 (red). The retinal-binding lysine, Lys296, is shown in black. A more detailed sequence alignment is provided in *Figure 1—figure supplement 2*. Residues are numbered according to bovine rhodopsin. (**C**) Absorption spectra in the dark of three *A. tenuis* opsins in the ASO-II group (Antho2a, Antho2c, and Antho2e). The absorption spectra were measured at 0°C in 140 mM NaCl at pH 6.5. The number in each graph shows the $\lambda_{max}$ value. (**D**) Results of the aequorin-based bioluminescent reporter assay for monitoring light-induced changes in Ca²⁺ in HEK293S cells expressing the same three opsins in the ASO-II group as in panel C. In each graph, luminescence values were normalized to the baseline. Black circles with error bars indicate the means ± SEMs (n=3) of the measured relative luminescence. Black arrowheads at time 0 indicate the timing of 1 min irradiation with green (495 nm; for Antho2a and Antho2c) or ultraviolet (UV) (395 nm; for Antho2e) light.

The online version of this article includes the following source data and figure supplement(s) for figure 1:

**Source data 1.** Raw absorbance values of purified pigments of Antho2a, Antho2c, and Antho2e recorded in the dark in the wavelength range of 250–750 nm.

**Source data 2.** Relative Ca²⁺ responses values (fold changes in luminescence above baseline levels) of wild types of Antho2a, Antho2c, and Antho2e.

**Figure supplement 1.** Maximum-likelihood (ML) tree of animal opsins, with non-opsin G protein-coupled receptors (GPCRs) included as an outgroup (a simplified version of the ML tree is shown in *Figure 1A*).

**Figure supplement 2.** Key residues in opsins of the anthozoan-specific opsin II (ASO-II) group and other animal opsins.

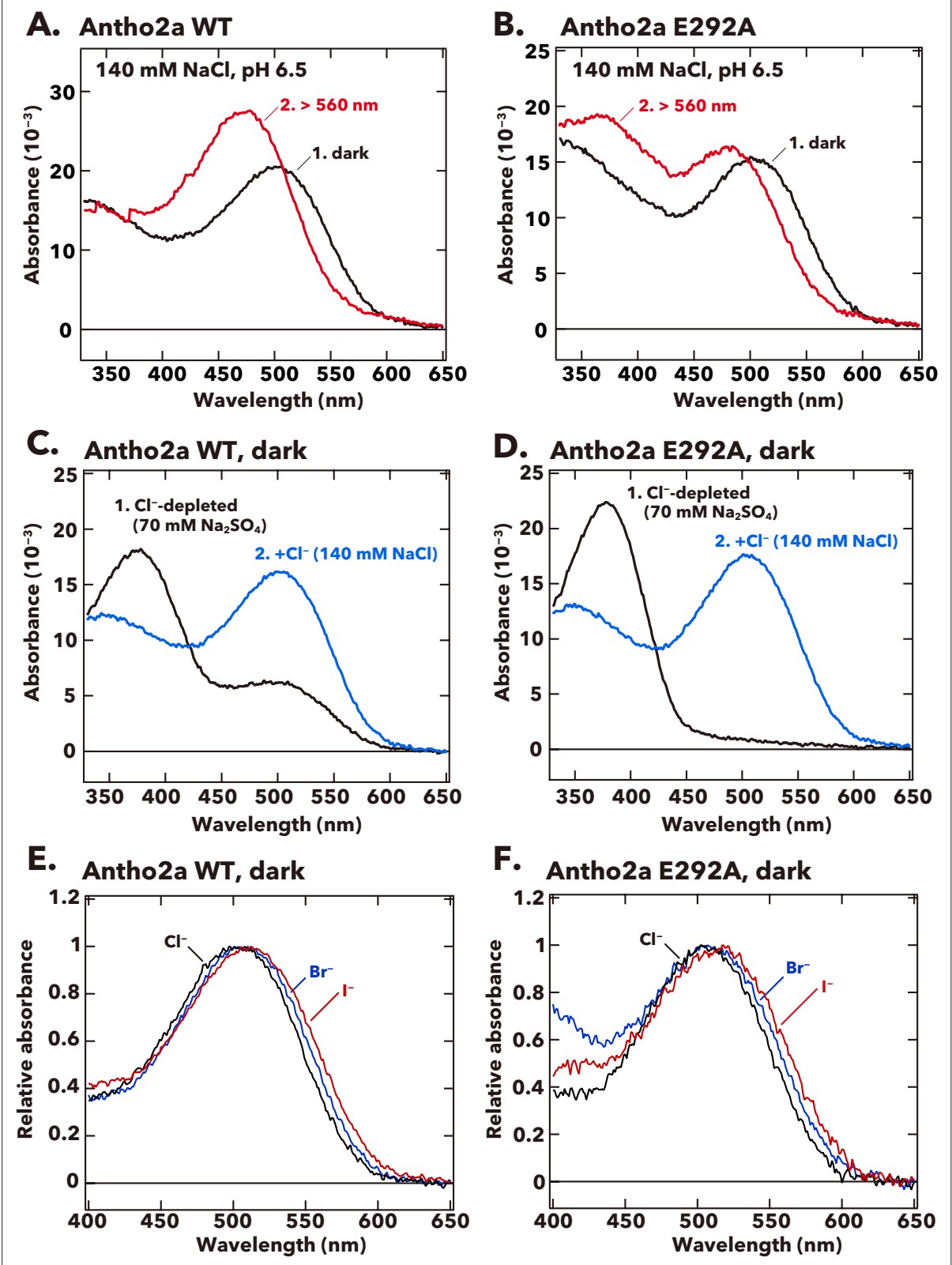

**Figure 2.** Absorption spectra of wild-type and the E292A mutant of *A. tenuis* Antho2a. (**A, B**) Absorption spectra of the dark state (curve 1, black) and the photoproduct (curve 2, red) of the wild-type (Antho2a WT, **A**) and the E292A mutant (Antho2a E292A, **B**) at 140 mM NaCl and pH 6.5. The samples were kept at 0°C during the spectroscopic measurements. (**C, D**) Absorption spectra of the dark state of Antho2a WT (**C**) and Antho2a E292A (**D**) prepared in Cl⁻-depleted conditions, before (curve 1, black) and after (curve 2, blue) adding Cl⁻ (see Materials and methods for details). In the

*Figure 2 continued on next page*

*Figure 2 continued*

Cl⁻-depleted condition, the pigments were solubilized in 70 mM $Na_2SO_4$, which reportedly does not access to the Cl⁻ binding site in the chicken red-sensitive cone visual pigment iodosin (*Shichida et al., 1990*) to moderate protein denaturation. (**E, F**) Effect of halide anions on the absorption spectra of wild-type Antho2a (**E**) and the Antho2a E292A mutant (**F**) at pH 6.5 and 0°C. The graphic shows the normalized absorption spectra of the pigments prepared in 140 mM NaCl (black curves), 140 mM NaBr (blue curves), and 140 mM NaI (red curves).

The online version of this article includes the following source data and figure supplement(s) for figure 2:

**Source data 1.** Raw absorbance values.

**Figure supplement 1.** Photo- and thermal reactions of wild-type Antho2a.

**Figure supplement 1—source data 1.** Raw absorbance values and raw HPLC data.

**Figure supplement 2.** Effect of pH on the absorption spectra of the dark states and photoproducts of Antho2a.

**Figure supplement 2—source data 1.** Raw absorbance values.

**Figure supplement 3.** Effect of pH and NaCl concentration on absorption spectra of the dark states and photoproducts of Antho2a.

**Figure supplement 3—source data 1.** Raw absorbance values.

also observed that the protonated photoproduct decayed more rapidly at pH 8.0 (*Figure 2—figure supplement 1H*) than at pH 6.5 (*Figure 2—figure supplement 1A, D, and E*). In contrast to the dark state, the photoproduct of the E292A mutant displayed two distinct absorption peaks in UV and visible light regions, at ~370 nm and 476 nm, respectively (*Figure 2B*, curve 2). This suggests that the E292A mutation causes UV-light absorption due to a deprotonated Schiff base in the photoproduct. Additionally, altering the pH modified the ratio of absorbance between the ~370 nm and 476 nm peaks in the E292A mutant (*Figure 2—figure supplement 2B*, curves 2), with the UV-peak to the visible light-peak ratio increasing at higher pH levels (pH 7.4, *Figure 2—figure supplement 2B*, curve 2). Conversely, the wild type did not exhibit an increase in UV absorbance under similar high pH conditions (pH 7.5, *Figure 2—figure supplement 2A*, curve 2). These results indicate that the Schiff base in the photoproduct of the Antho2a E292A mutant has a lower acid dissociation constant ($pK_a$) than that of the wild type, suggesting that Glu292 acts as the counterion in the photoproduct of Antho2a.

We then further explored the nature of the counterion in the dark state of Antho2a. Previous studies have shown that in the bovine rhodopsin E113A and E113Q mutants, as well as in the retinochrome E181Q mutant (referred to as 'counterion-less' mutants), halide ions like Cl⁻ can act as 'surrogate' counterions to stabilize the proton on the Schiff base. Consequently, these counterion-less mutants can still absorb visible light in the presence of Cl⁻ (*Nathans, 1990*; *Sakmar et al., 1991*; *Terakita et al., 2000*). To assess the potential role of Cl⁻ as a surrogate counterion in the dark state of the Antho2a E292A mutant, we performed spectroscopic analyses under Cl⁻-depleted conditions (*Figure 2C and D*). We observed that the $\lambda_{max}$ of the E292A mutant shifted to the UV region (*Figure 2D*, curve 1). Unexpectedly, a similar shift in absorption to the UV region was also observed in the wild type under the Cl⁻-depleted condition (*Figure 2C*, curve 1). These results indicate that, in the absence of Cl⁻, the Schiff base in both the wild-type and the E292A dark states becomes deprotonated. The subsequent addition of Cl⁻ (final concentration: 140 mM NaCl) restored clear absorbance in the visible light region (*Figure 2C and D*, curves 2), showing that Cl⁻ facilitates the protonation of the Schiff base of the dark state even in the wild type. In contrast, the photoproduct of the wild type exhibited no significant change in the ratio of UV to visible-light absorption peaks at pH 6.5 across NaCl concentrations from 0.28 mM to 800 mM (*Figure 2—figure supplement 3*). The photoproduct of the wild type consistently absorbed visible light under these NaCl conditions (curves 2 in *Figure 2—figure supplement 3A and B*), suggesting that Cl⁻ has little impact on the Schiff base $pK_a$ in the photoproduct of wild-type Antho2a. However, the photoproduct of the E292A mutant exhibited a pH-dependent shift in the ratio of UV to visible-light absorption between pH 4.8 and pH 7.6, even at 800 mM NaCl, where the dark state predominantly absorbed visible light (*Figure 2—figure supplement 3C*). This further supports that Glu292 serves as the counterion in the photoproduct of Antho2a.

## Effect of halide anions on $\lambda_{max}$ values of the dark state of Antho2a

To obtain further evidence supporting the Cl⁻ counterion in the dark state of Antho2a, we examined the impact of different halide anions on the absorption spectrum in the dark state of Antho2a, as observed in the bovine rhodopsin counterion-less mutant (*Nathans, 1990*; *Sakmar et al., 1991*).

Antho2a readily absorbed visible light in the presence of bromide ion (Br⁻) and iodide ion (I⁻), as well as Cl⁻, and the $\lambda_{max}$ of wild-type Antho2a shifted depending on the halide solutions (503 nm in 140 mM NaCl; 506 nm in 140 mM NaBr; 511 nm in 140 mM NaI solutions; *Figure 2E*). The E292A mutant showed a similar shift in $\lambda_{max}$ (505 nm in 140 mM NaCl; 507 nm in 140 mM NaBr; 517 nm in 140 mM NaI solutions; *Figure 2F*).

## Effect of Cl⁻ concentration on the p$K_a$ of the protonated Schiff base of Antho2a

To further investigate the influence of Cl⁻ on the protonation state of the Schiff base in the dark state of Antho2a, we estimated the p$K_a$ of the Schiff base by measuring the pH-dependent changes in the absorption spectra of Antho2a at different Cl⁻ concentrations. The pH-dependent equilibrium between the visible (protonated Schiff base) and UV (deprotonated Schiff base) forms revealed that their ratio changes with Cl⁻ concentration (*Figure 3—figure supplement 1A–E*). A plot of the changes in absorbance at $\lambda_{max}$ against pH (*Figure 3A*) shows that in wild-type Antho2a, the p$K_a$ of the protonated Schiff base increases with higher Cl⁻ concentrations (7.3 at 0.28 mM NaCl, 8.0 at 2.8 mM, 8.8 at 28 mM, 8.8 at 140 mM, and 9.0 at 500 mM). We failed to determine the p$K_a$ at 0 mM NaCl, as the observed $\lambda_{max}$ in acidic conditions (pH <6.5) was shorter than expected in Antho2a (503 nm), suggesting that a normal pigment was not produced under these conditions (*Figure 3—figure supplement 2A*). Similarly, the Cl⁻ concentration also affected the p$K_a$ of the protonated Schiff base in the E292A mutant (6.1 at 2.8 mM NaCl, 6.8 at 28 mM, 7.7 at 140 mM, and 8.9 at 500 mM) (*Figure 3B*; *Figure 3—figure supplement 3*). At 0 mM NaCl, the E292A mutant showed no visible light absorption, even under the most acidic conditions (pH 4.7), preventing the determination of its p$K_a$ (*Figure 3—figure supplement 2B*). Notably, at low Cl⁻ concentrations (2.8 mM NaCl), the wild type exhibited a higher p$K_a$ than the E292A mutant (8.0 and 6.1, respectively). However, at 500 mM NaCl, the p$K_a$ of the E292A mutant and wild type were comparable (9.0 and 8.9, respectively; *Figure 3A and B*). These results suggest that Cl⁻, rather than Glu292, serves as the counterion in the dark state of Antho2a, while Glu292 facilitates the protonation of the Schiff base by the Cl⁻ counterion.

## Binding affinity of Cl⁻ to wild-type Antho2a and the E292A mutant

To evaluate the Cl⁻ binding affinities of both wild-type Antho2a and the E292A mutant, we measured changes in their absorption spectra by gradually increasing Cl⁻ concentrations at pH 6.5 and estimated the Cl⁻ dissociation coefficients ($K_d$). The relative absorbance in the visible region increased with higher Cl⁻ concentrations both in the wild type and in the E292A mutant (*Figure 3—figure supplement 4A and B*). By fitting the Hill equation to the experimental data (*Figure 3C*), the dissociation constants ($K_d$) of Cl⁻ were determined to be 0.079±0.010 mM for the wild-type Antho2a and 12.7±0.519 mM for the E292A mutant. This significant increase in the $K_d$ value for the E292A mutant suggests that the Cl⁻ binding affinity is considerably reduced due to the mutation. Consequently, we suggest that while Glu292 does not act as a direct counterion, it plays a crucial role in facilitating Cl⁻ binding to Antho2a.

## Structural modeling and QM/MM calculations of the dark state of Antho2a

To gain a deeper understanding of the environment surrounding the retinylidene Schiff base in the dark state of Antho2a, we performed QM/MM-based structural modeling of both the wild-type Antho2a (with Glu292 either neutral or negatively charged) and the E292A mutant. The QM/MM geometry optimization positioned the Cl⁻ ion close to the Schiff base (~3 Å) and near Glu292 (~4.7 Å), with Glu292 itself located in proximity to the Schiff base (~3.3 Å) (*Figure 4B*). The chloride ion is also coordinated by two water molecules and the backbone of Cys187 which is part of a conserved disulfide bridge (*Figure 1—figure supplement 2*). The retinylidene Schiff base region also includes polar (Ser186, Tyr91) and nonpolar (Ala94, Leu113) residues (*Figure 4*). To validate these models, we calculated the QM/MM vertical excitation energies of the ground state geometries (*Table 1*).

For wild-type Antho2a with a protonated neutral Glu292, the calculated $\lambda_{max}$ using the CAM-B3LYP/cc-pVTZ level of theory was 503 nm (*Figure 4B*), in good agreement with the experimentally observed value (503 nm; *Figure 2A*). In contrast, the $\lambda_{max}$ calculated with a deprotonated negatively charged Glu292 was blue-shifted to 415 nm (*Figure 4C*), deviating significantly from the experimental

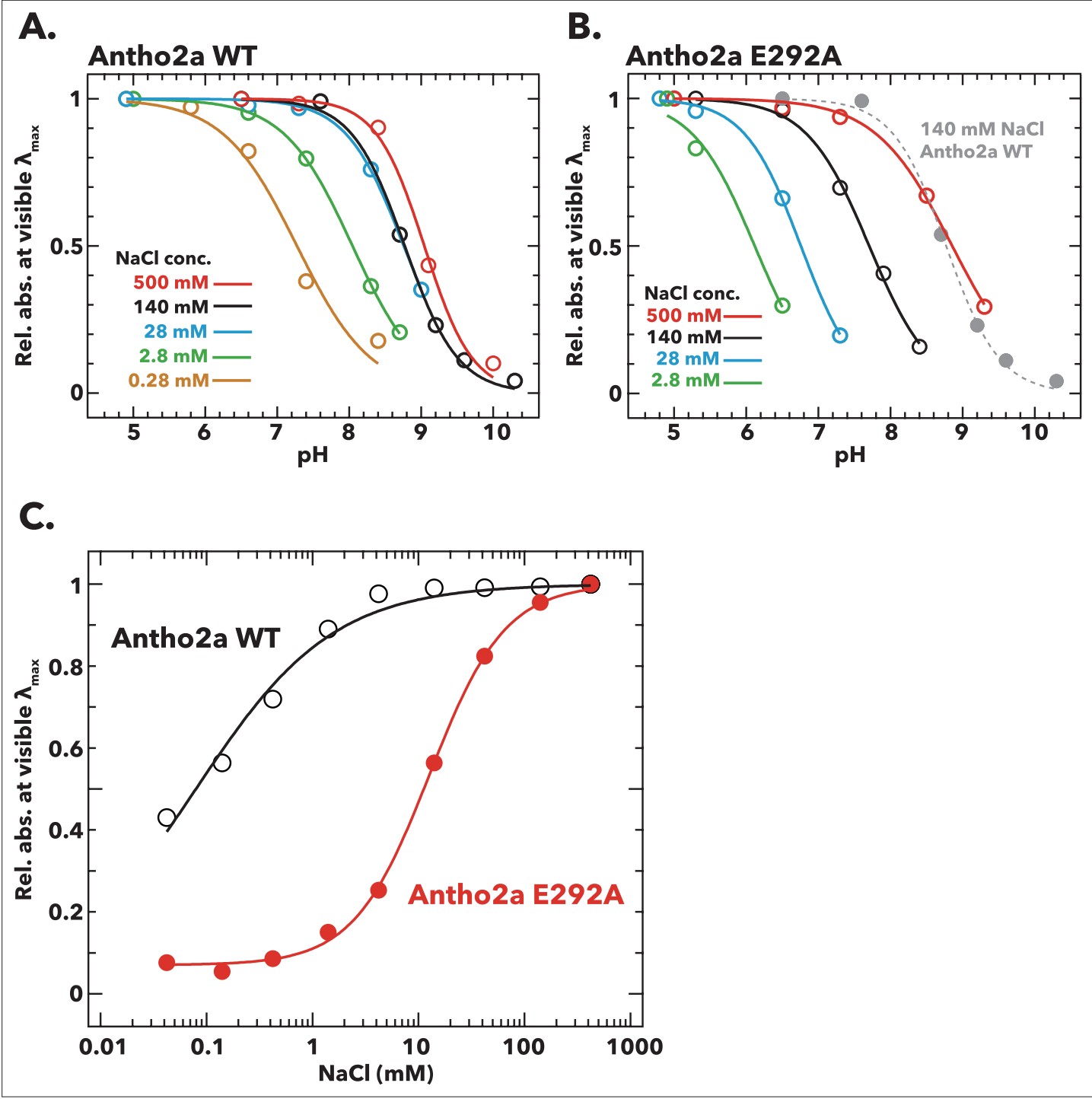

**Figure 3.** Effects of pH and Cl⁻ concentration on the absorption spectra of the dark states of wild-type Antho2a and the Antho2a E292A mutant. (**A, B**) Changes in the absorbance at $\lambda_{max}$ as a function of pH for (**A**) wild-type Antho2a and (**B**) the E292A mutant at different Cl⁻ concentrations. The absorbance values at 'visible $\lambda_{max}$' (mean absorbance at 503±5 nm for the wild type and 505±5 nm for the E292A mutant, respectively) were normalized for each Cl⁻ concentration to those at the lowest pH, in which the Schiff base is assumed to be fully protonated ('Rel. abs. at visible $\lambda_{max}$' in the y-axes). Solid and dashed lines represent sigmoid fits to the experimental data for each Cl⁻ concentration (indicated by different colors). The pH-dependent change of wild-type Antho2a at 140 mM NaCl is also shown in panel B (dotted gray line). The full absorption spectra used to generate these plots are provided in *Figure 3—figure supplement 1* (for wild-type Antho2a) and *Figure 3—figure supplement 3* (for the E292A mutant). (**C**) Changes in the absorbance at $\lambda_{max}$ for wild-type Antho2a (black open circles) and the E292A mutant (red solid circles) as a function of Cl⁻ concentration. The absorbance values at visible $\lambda_{max}$ were normalized to those at 500 mM NaCl for both the wild type and the E292A mutant. The lines in the graph were

*Figure 3 continued on next page*

*Figure 3 continued*

generated by fitting the Hill equation to the experimental data. The full absorption spectra used to generate these plots are provided in *Figure 3— figure supplement 4*.

The online version of this article includes the following source data and figure supplement(s) for figure 3:

**Source data 1.** Summary of mean relative absorbance values at $\lambda_{max}$ (±5 nm) at different pH and NaCl concentrations.

**Figure supplement 1.** pH-dependent changes in the absorption spectra of Antho2a wild type (WT) at (**A**) 0.28 mM, (**B**) 2.8 mM, (**C**) 28 mM, (**D**) 140 mM, and (**E**) 500 mM Cl⁻ at 0°C.

**Figure supplement 1—source data 1.** Raw absorbance values.

**Figure supplement 2.** pH-dependent changes in the absorption spectra of Antho2a wild type (WT) and Antho2a E292A at 0 mM NaCl (containing 70 mM $Na_2SO_4$) at 0°C.

**Figure supplement 2—source data 1.** Raw absorbance values.

**Figure supplement 3.** pH-dependent changes in the absorption spectra of Antho2a E292A at (**A**) 2.8 mM, (**B**) 28 mM, (**C**) 140 mM, and (**D**) 500 mM Cl⁻ at 0°C.

**Figure supplement 3—source data 1.** Raw absorbance values.

**Figure supplement 4.** Absorption spectra of (**A**) wild-type Antho2a and (**B**) the Antho2a E292A mutant under different Cl⁻ concentrations at pH 6.5 and 0°C.

**Figure supplement 4—source data 1.** Raw absorbance values.

value. Finally, the calculated $\lambda_{max}$ for the E292A mutant was 499 nm (*Figure 4D*), also in agreement with the experimental value (505 nm). To further substantiate these findings, we recalculated the excitation energies using the RI-ADC(2)/cc-pVTZ method. Although these $\lambda_{max}$ values are blue-shifted compared to those calculated with the CAM-B3LYP method, they followed a similar trend. Both of these computational methods have previously been employed to accurately calculate the excitation energies of rhodopsins (*Church et al., 2021*). These results strongly suggest that in the dark state of Antho2a, Glu292 is protonated and neutral at pH 6.5, and therefore, it does not function as the counterion.

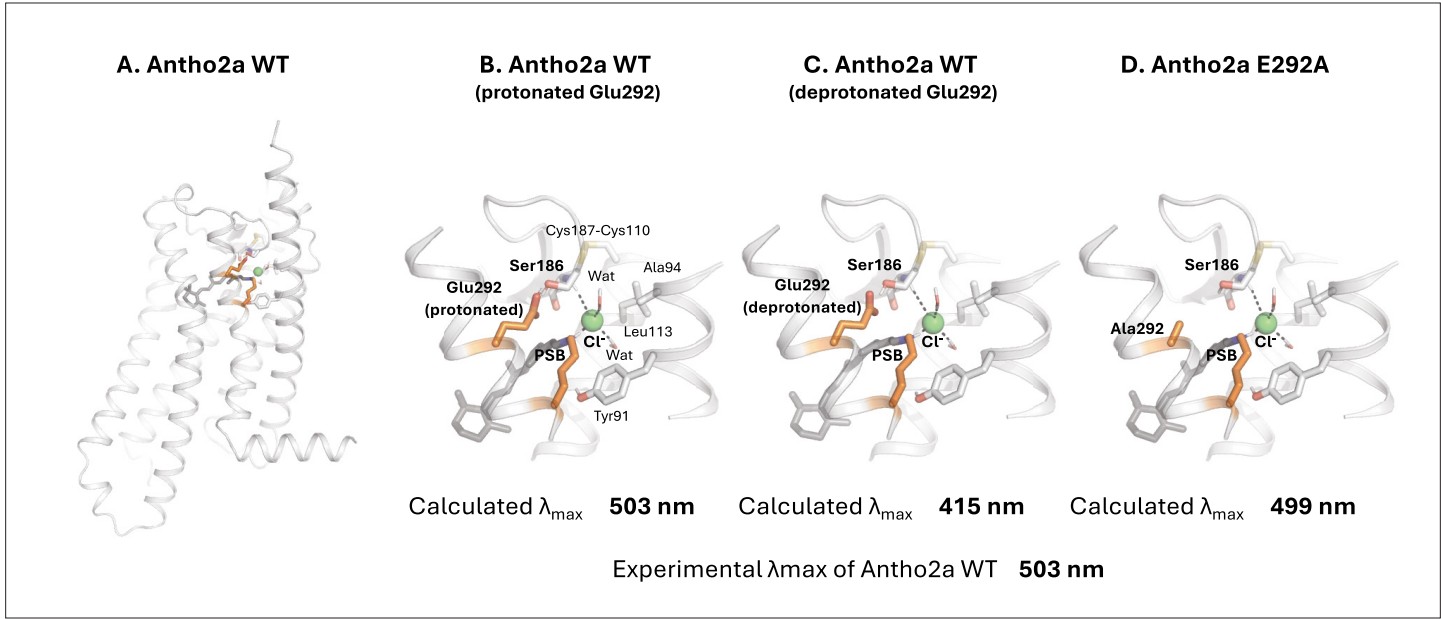

**Figure 4.** Quantum mechanics/molecular mechanics (QM/MM) structural model of wild-type Antho2a in the dark state (**A**) and detailed views of the retinal binding pocket with a protonated (neutral) Glu292 (**B**), a deprotonated (negatively charged) Glu292 (**C**), and the E292A mutant (**D**). The retinal protonated Schiff base (PSB) and the binding pocket residues are shown as sticks (including polar hydrogens) and the Cl⁻ ion as a sphere with its coordination shown as dashes. 'Wat' indicates a water molecule. Residues in the QM region are marked in bold.

**Table 1.** Vertical excitation energies ($\Delta E_{calc}$) and oscillator strengths ($f$) computed by quantum mechanics/molecular mechanics (QM/MM) calculations using different QM methods with the cc-pVTZ basis set.

| | sTD-DFT CAM-B3LYP | | ADC(2) | |
|---|---|---|---|---|
| **QM Region** | $\Delta E_{calc}$ nm (eV) | $f$ | $\Delta E_{calc}$ nm (eV) | $f$ |
| RET +Lyr296+Cl + Ser186+**Glu292 (deprotonated)** | 415 (2.99) | 1.54 | 374 (3.32) | 1.64 |
| RET +Lyr296+Cl + Ser186+**Glu292 (protonated)** | 503 (2.47) | 1.19 | 416 (2.98) | 1.43 |
| RET +Lyr296+Cl + Ser186+**Ala292** | 499 (2.49) | 1.19 | 426 (2.91) | 1.36 |

## Effect of the Glu292 mutation on the function of the photoproduct

The spectroscopy data indicate that Glu292 is involved in stabilizing the protonated Schiff base by facilitating $Cl^-$ binding in the dark state and also serves as a counterion in the photoproduct. This suggests that Glu292 significantly contributes to the visible light absorption of Antho2a. To explore additional roles of Glu292 in Antho2a, we measured the light-induced $Ca^{2+}$ response in cultured cells expressing wild-type Antho2a or the E292A mutant. Notably, cells expressing wild-type Antho2a showed an ~30-fold increase in $Ca^{2+}$ levels upon light irradiation (**Figure 5**, solid black circles), whereas cells expressing the Antho2a E292A mutant showed a smaller $Ca^{2+}$ elevation (<5-fold increase) (**Figure 5**,

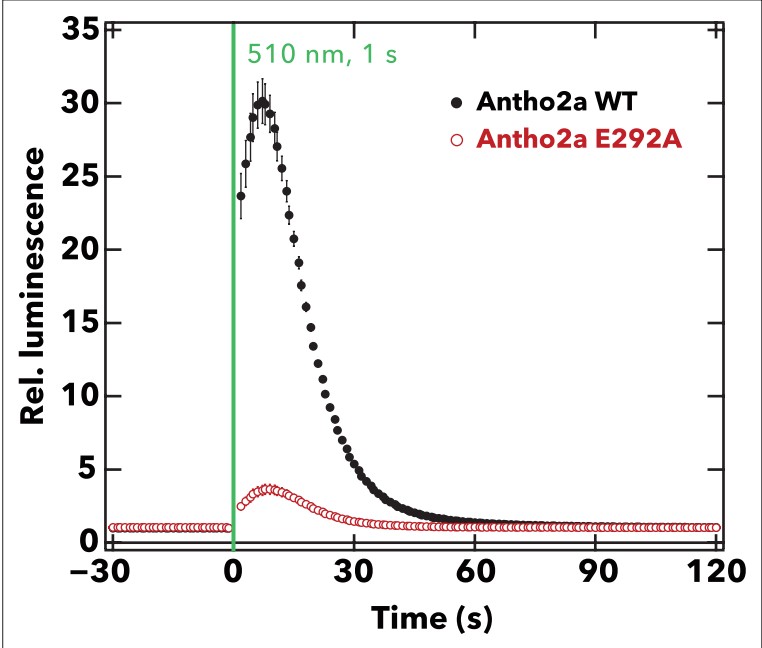

**Figure 5.** Comparison of the light-evoked intracellular $Ca^{2+}$ levels between wild-type Antho2a and the E292A mutant. The graph shows the mean ± SEM (n=4) of the measured relative luminescence values (luminescence values normalized to the baseline) for wild-type Antho2a (black) and the E292A mutant (red) at pH 7.0. The green vertical line indicates the time of cell illumination with green light (510 nm, for 1 s, $1.65 \times 10^{15}$ photons/cm²/s).

The online version of this article includes the following source data, source code, and figure supplement(s) for figure 5:

**Source data 1.** Mean and SEM values of relative Ca2+ 1024 responses (fold changes in luminescence above baseline levels) of wild type and E292A Antho2a.

**Figure supplement 1—source data 1.** Summary of relative expression values and normalized $Ca^{2+}$ response values for wild type and E292A mutant of Antho2a.

**Figure supplement 1—source code 1.** R code for analyzing $Ca^{2+}$ response data in *Figure 5—figure supplement 1—source data 1*.

**Figure supplement 1.** Expression levels and normalized $Ca^{2+}$ responses of wild-type and E292A Antho2a.

red open circles). This indicates that the peak $Ca^{2+}$ response in cells expressing wild-type Antho2a was approximately nine times greater than in cells expressing the E292A mutant. This result, along with the crucial role of Glu292 in $Cl^-$ binding in the dark state and as a counterion in the photoproduct, suggests that Glu292 also plays a role in G protein activation.

## $Cl^-$-dependent changes in the absorption spectra of the dark states of Antho2c and Antho2e

We tested whether $Cl^-$ concentration affects the $pK_a$ of the Schiff base in another visible light-sensitive opsin, Antho2c ($\lambda_{max}$ = 450 nm, *Figure 1C*). The pH-dependent equilibrium between UV- and visible-light absorbing forms was clearly observed at 0 mM or 0.093 mM NaCl, but not at 9.3 mM NaCl, where Antho2c stably absorbed visible light across the measured pH range (pH 4.8–7.2, *Figure 6A–C*). Also, the ratio of UV to visible-light absorption increased with higher $Cl^-$ concentrations at pH 6.5 (*Figure 6D*). These results demonstrate that $Cl^-$ serves as a counterion in the dark state of Antho2c, as it does in Antho2a. In contrast, wild-type Antho2e continues to absorb UV light even at 1 M NaCl (*Figure 6E*). Notably, Antho2e has an arginine at position 113, which corresponds to the counterion position in vertebrate visual opsins (*Figure 1—figure supplement 2*). When this arginine is mutated to alanine (R113A), the mutant becomes sensitive to visible light ($\lambda_{max}$ = ~420 nm) in the presence of $Cl^-$ (*Figure 6F*), suggesting that $Cl^-$ can serve as the counterion in the R113A mutant.

## Discussion

In this study, we reveal for the first time the spectral properties of opsins in the ASO-II group from the coral *A. tenuis*, showing that their sensitivity spans from UV to visible light. Opsins in this group have a highly conserved Glu292 residue near the Schiff base, which can potentially stabilize the proton on the Schiff base. Indeed, our results show that the $pK_a$ of the protonated Schiff base in the photo-product ($\lambda_{max}$ = 476 nm) of Antho2a is altered when Glu292 is substituted with alanine (*Figure 2A and B*; *Figure 2—figure supplement 2*), suggesting that Glu292 serves as the counterion of the photoproduct. Conversely, the dark state of Antho2a ($\lambda_{max}$ = 503 nm) exhibits robust visible light absorption only in the presence of $Cl^-$ at physiological pH, and the $pK_a$ of the protonated Schiff base changes with $Cl^-$ concentration in both wild-type Antho2a and the E292A mutant. Furthermore, the $pK_a$ for wild type and the E292A mutant is comparable in the presence of sufficient $Cl^-$ (500 mM NaCl, *Figure 3A and B*), supporting the conclusion that $Cl^-$, and not Glu292, acts as the counterion of the dark state of *A. tenuis* Antho2a. We found that the type of halide anions in the solution has a small but noticeable effect on the $\lambda_{max}$ values of the dark state of Antho2a. This is consistent with the effect observed in a counterion-less mutant of bovine rhodopsin, in which halide ions serve as surrogate counterions (*Nathans, 1990*; *Sakmar et al., 1991*). Similarly, our results align with earlier observations that the $\lambda_{max}$ of a retinylidene Schiff base in solution increases with the ionic radius of halides acting as hydrogen bond acceptors (i.e. $I^- > Br^- > Cl^-$) (*Blatz et al., 1972*). In contrast, the $\lambda_{max}$ of halorhodopsin from *Natronobacterium pharaonic* does not clearly correlate with halide ionic radius (*Scharf and Engelhard, 1994*), as the halide ion in this case is not a hydrogen-bonding acceptor of the protonated Schiff base (*Kouyama et al., 2010*; *Mizuno et al., 2018*). Altogether, these findings support our hypothesis that in Antho2a, a solute halide ion forms a hydrogen bond with the Schiff base, thereby serving as the counterion in the dark state. Moreover, QM/MM calculations for the dark state of Antho2a suggest that Glu292 is protonated and neutral, further supporting the hypothesis that Glu292 does not serve as the counterion in the dark state. However, unlike the dark state, $Cl^-$ has little to no effect on the visible light absorption of the photoproduct (*Figure 2—figure supplement 3*). Therefore, we conclude that $Cl^-$ and Glu292, respectively, act as counterions for the protonated Schiff base of the dark state and photoproduct of Antho2a. This represents a unique example of counterion switching from exogenous anion to a specific amino acid residue upon light irradiation (*Figure 7*).

Our spectroscopic data also showed that the other visual light-sensitive opsin, Antho2c, exhibited the $Cl^-$ dependency on the Schiff base $pK_a$ of the dark state, which suggested that opsins in the ASO-II group may share a spectral tuning mechanism based on the $Cl^-$ counterion. Interestingly, Antho2e had an arginine at position 113, and when it was mutated to alanine (R113A), the mutant showed sensitivity to visible light in the presence of $Cl^-$ (*Figure 6F*). We hypothesize that the positive charge

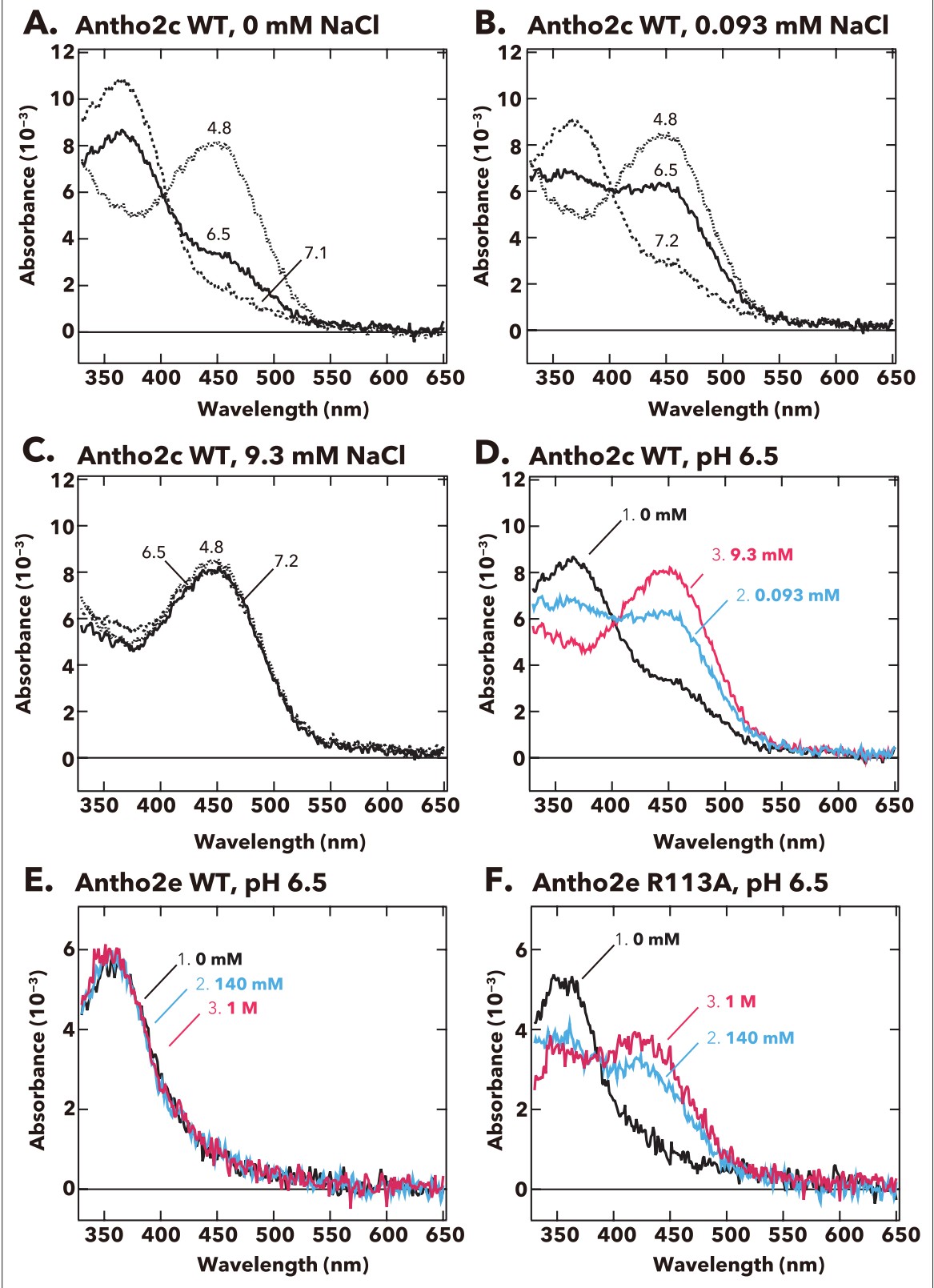

**Figure 6.** pH-dependent changes in the absorption spectra of Antho2c and Antho2e at different Cl⁻ concentrations at 0°C. (A–C) Absorption spectra of purified wild-type Antho2c pigment at (**A**) 0 mM, (**B**) 0.093 mM, and (**C**) 9.3 mM NaCl concentrations. The corresponding pH values are indicated on each curve in the graphs. (**D**) Summary of the spectral changes for wild-type Antho2c across different Cl⁻ concentrations at neutral pH (pH 6.5). (**E, F**)

*Figure 6 continued on next page*

*Figure 6 continued*

Absorption spectra of (**E**) wild-type Antho2e (Antho2e WT) and (**F**) its R113A mutant (Antho2e R113A) at different Cl⁻ concentrations at pH 6.5 at 0°C. Each color indicates a different Cl⁻ concentration.

The online version of this article includes the following source data for figure 6:

**Source data 1.** Raw absorbance values.

of Arg113 disturbs interaction between the Cl⁻ and the protonated Schiff base or that it completely inhibits Cl⁻ binding, rendering Antho2e UV-sensitive.

It is noteworthy that although Cl⁻ has been reported to serve as a surrogate counterion in 'counterion-less' mutants of animal opsins (such as E113Q in bovine rhodopsin and E181Q in retinochrome) (*Nathans, 1990*; *Sakmar et al., 1991*; *Terakita et al., 2000*), Antho2a is, to our knowledge, the first example in a wild-type animal opsin that employs Cl⁻ as a counterion. Interestingly, within the microbial rhodopsin family, heliorhodopsin (TaHeR) incorporates a Cl⁻ into the Schiff base region under high Cl⁻ concentrations at pH 4.5. This Cl⁻ stabilizes the protonated retinal Schiff base when its primary counterion, E108, is neutralized (*Besaw et al., 2022*). The observed 18 nm red shift at low pH is consistent with E108 protonation. The TaHeR E108A mutant shows the same $\lambda_{max}$ under high

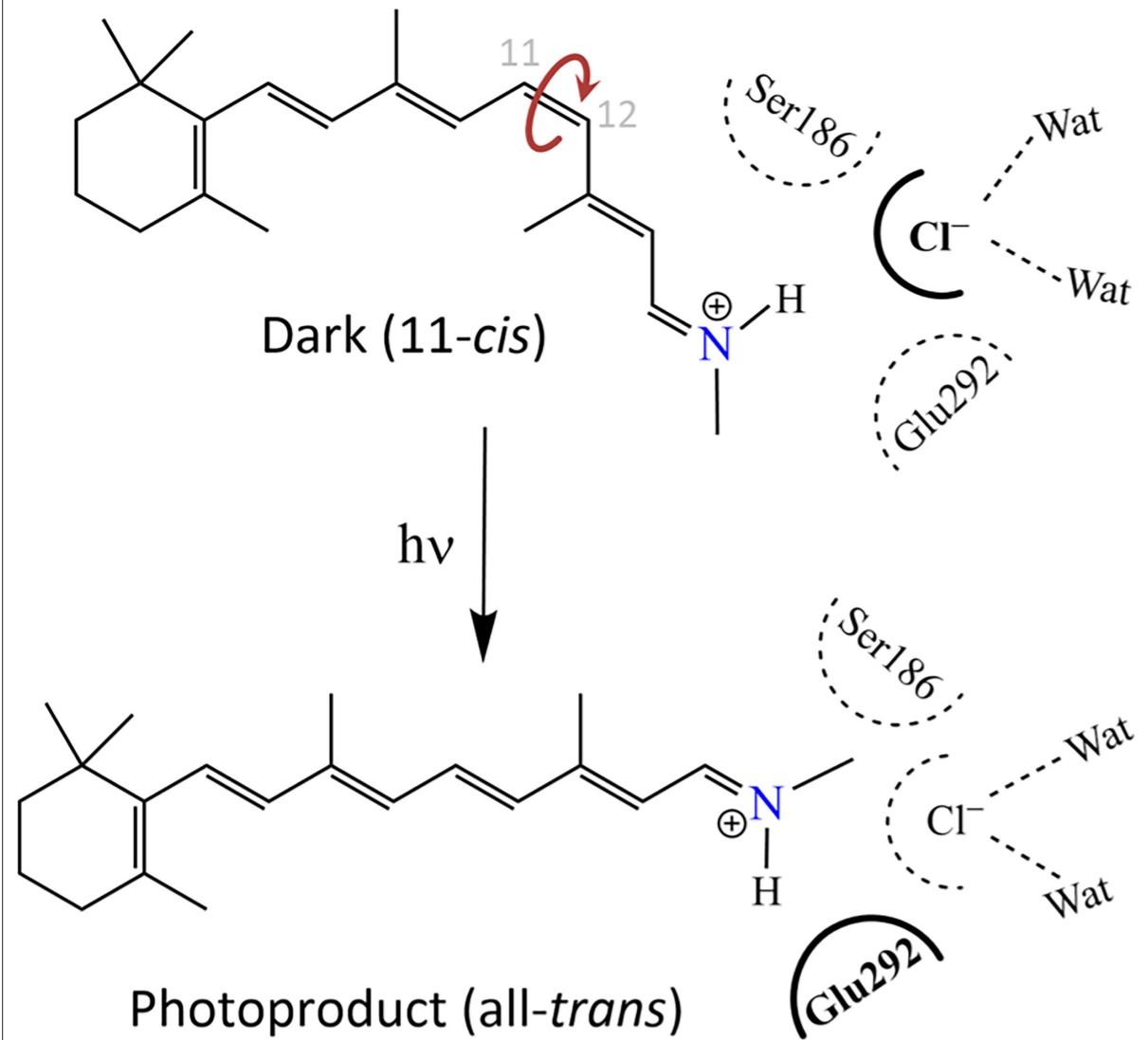

**Figure 7.** Schematic drawing of the environment of the protonated Schiff base depicting the counterion switch from a chloride ion (top) to Glu292 (bottom) upon light activation.

$Cl^-$ concentrations, further supporting the role of $Cl^-$ as a counterion. At pH 8, the Schiff base proton is stabilized by the negatively charged E108 (*Shihoya et al., 2019*).

The E292A mutation in Antho2a drastically decreases the $Cl^-$ binding affinity in the dark state at pH 6.5 ($K_d$: Antho2a WT = 0.079 mM, Antho2a E292A=12.7 mM). Based on these results, supported by our QM/MM calculations of the dark state, we hypothesize that protonated Glu292 not only serves as a counterion of the photoproduct but also constitutes part of the $Cl^-$ binding site in the dark state of Antho2a. In $Cl^-$-pumping microbial rhodopsins, where $Cl^-$ serves as a counterion, the $Cl^-$ binding site near the Schiff base is typically formed by hydrogen-bonding network between $Cl^-$, the protonated Schiff base, and key amino acids such as serine, threonine, and glutamic/aspartic acid, and water molecules (*Besaw et al., 2020*; *Hosaka et al., 2016*; *Kolbe et al., 2000*). Additionally, the bovine rhodopsin double mutant E113Q/A292E exhibits a higher sensitivity to hydroxylamine than the wild type, indicating instability of the Schiff base and increased solvent accessibility (*Tsutsui and Shichida, 2010*). These observations suggest that Glu292 in Antho2a may also facilitate the accessibility of $Cl^-$ ions to the Schiff base environment. To determine the precise nature of the $Cl^-$ binding site and clarify the roles of Glu292 and other residues in the $Cl^-$-dependent counterion system of Antho2a, detailed spectroscopic and structural experiments will be necessary.

We also found that cells expressing the Antho2a E292A mutant show a lower $Ca^{2+}$ elevation upon light stimulation compared to the wild type (*Figure 5*). The relative expression level of the E292A mutant of Antho2a was approximately 0.81 of the wild type (set as 1), as determined by comparing absorbances at $\lambda_{max}$ for both pigments expressed and purified under identical conditions. Additionally, the fraction of protonated pigment relative to the wild type (set as 1 at pH 6.5) was estimated to be 0.94 for the E292A mutant at pH 6.5, and 0.99 and 0.84 for the wild type and the E292A mutant at pH 7.0, respectively (*Figure 3A and B*). Since pH 7.0 corresponds to the conditions used in the live-cell $Ca^{2+}$ assays, the effective amount of protonated pigment for the E292A mutant was approximately 73% of the wild type. Nevertheless, even after normalization for these differences, the $Ca^{2+}$ response amplitude of the E292A mutant remained significantly lower (~17% of wild type, compared to the observed 12% prior to normalization; *Figure 5*, *Figure 5—figure supplement 1*). These observations suggest that Glu292 serves not only as a counterion in the photoproduct but also plays an allosteric role in influencing G protein activation. It has been reported that the introduction of Glu or Asp at position 292 can affect various molecular properties of opsins. For instance, the A292E mutation in human rhodopsin has been shown to result in constitutively active apoproteins, leading to congenital night blindness (*Dryja et al., 1993*; *Jin et al., 2003*). These data suggest that Glu292 has the potential to influence functional properties, such as G protein activation. Several studies have reported that the counterion can affect diverse properties of opsins beyond their absorption spectra. For instance, in bovine rhodopsin, Glu113 is involved not only in visible light absorption in the dark state but also in the efficient activation of G proteins by the photoproduct (*Terakita et al., 2004*). It has been suggested that these pleiotropic functions of the counterion have been achieved through evolutionary modifications of the protein structure once the counterion was acquired (*Terakita et al., 2004*). This concept may similarly apply to Antho2a.

One of the notable features of the opsins in the ASO-II group that use $Cl^-$ as a counterion is the relatively low p$K_a$ of the Schiff base (e.g. ~9.0 for Antho2a in the dark state, *Figure 3A*) compared to other animal opsins with a 'regular' amino acid counterion (e.g. >16 for bovine rhodopsin, with Glu113 as a counterion [*Steinberg et al., 1993*] or 11 for jumping spider Rh1, with Glu181 as a counterion [*Nagata et al., 2019*]). We hypothesize that opsins in the ASO-II group, with lower Schiff base p$K_a$, may change their spectral sensitivity and G protein activation profile within the physiological pH range of corals. For instance, these opsins could exhibit decreased sensitivity to visible and increased sensitivity to UV light under alkaline pH conditions. The extra- and intracellular pH environments in symbiotic cnidarians such as corals are spatially and temporally variable due to the photosynthesis of symbiotic algae (*Barott et al., 2017*). For example, pH values in the cytosol of algae-hosting cells (a group of endodermal cells with algal occupancy) in corals reportedly increase by approximately 0.5 pH units upon light treatment (*Venn et al., 2009*). The pH values in the gastrovascular cavity (coelenteron), which is in contact with endodermal cells, range from 6.6 to 8.5 (*Agostini et al., 2012*; *Al-Horani et al., 2003*; *Cai et al., 2016*) and increase in the presence of light, presumably due to the consumption of $CO_2$ by photosynthesis (*Barott et al., 2017*). Additionally, alkalinization in the extracellular region of calcifying cells reportedly increases ~1.0 pH units from dark to light conditions, reaching

levels above pH 9.0 (*Al-Horani et al., 2003*). These pH changes, generally an increase due to photosynthetic activity, may result in variable light sensitivity of the 'low p$K_a$' opsins in the ASO-II group. Recent studies have reported that members of the ASO-II group may be associated with the symbiotic relationship between host anthozoans and symbiotic algae. For example, using the symbiotic sea anemone *Exaiptasia diaphana*, *Gornik et al., 2021*, showed that the mRNA levels of several opsins in the ASO-II group were higher in symbiotic adults than in apo-symbiotic ones (*Gornik et al., 2021*). In sea anemone, it has also been reported that behavioral responses to light differ between symbiotic and apo-symbiotic individuals (*Foo et al., 2020*; *Kishimoto et al., 2023*), with these responses potentially being driven by the activation of ASO-II opsins that are upregulated in the presence of symbiotic algae. Moreover, single-cell RNA-seq analysis in the reef-building coral *Stylophora pistillata* has revealed that an opsin belonging to the ASO-II group is specifically expressed in algae-hosting cells (a group of endodermal cells with 50% algal occupancy) (*Levy et al., 2021*). Although further studies are needed, we suggest that the unique use of Cl$^-$ as the counterion in opsins of the ASO-II group, rather than a negatively charged amino acid, may be associated with their pH-sensitive light response and, ultimately, to their role in photosynthesis-related functions in symbiotic cnidarians.

It is widely accepted that opsins have evolved from a non-opsin GPCR (*Feuda et al., 2012*). Our finding of the native chloride counterion in opsins shows a possibility that in the first stage of the evolutionary process, the 'primitive' opsins having Lys296 but no counterion residue could absorb not only UV light but also a wide range of wavelengths that extends into the visible region by embracing chloride ions. Namely, the chloride counterion system might have been a preliminary step in the evolution of amino acid counterions in animal opsins. Further empirical and bioinformatic studies are required for disentangling the evolutionary trajectories of the Schiff base-counterion system, including the chloride counterion.

# Materials and methods

## Key resources table

| Reagent type (species) or resource | Designation | Source or reference | Identifiers | Additional information |
|---|---|---|---|---|
| Gene (*Acropora tenuis*) | Antho2a | This study | GenBank: LC844932 | The sequence information is available from NCBI GenBank. |
| Gene (*Acropora tenuis*) | Antho2c | This study | GenBank: LC844934 | The sequence information is available from NCBI GenBank. |
| Gene (*Acropora tenuis*) | Antho2e | This study | GenBank: LC844936 | The sequence information is available from NCBI GenBank. |
| Recombinant DNA reagent | pUSRα-Antho2a_1D4 | This paper | | The coding sequence (CDS) of Antho2a was tagged with rho 1D4 epitope and inserted into the multicloning site of pUSRα vector (see Materials and methods section). Available from Akihisa Terakita lab. |
| Recombinant DNA reagent | pMT-Antho2a_1D4 | This study | | The CDS of Antho2a was tagged with rho 1D4 epitope and inserted into the multicloning site of pMT vector (see Materials and methods section). Available from Akihisa Terakita lab. |
| Recombinant DNA reagent | pUSRα-Antho2c_1D4 | This study | | The CDS of Antho2c was tagged with rho 1D4 epitope and inserted into the multicloning site of pUSRα vector (see Materials and methods section). Available from Akihisa Terakita lab. |
| Recombinant DNA reagent | pMT-Antho2c_1D4 | This study | | The CDS of Antho2c was tagged with rho 1D4 epitope and inserted into the multicloning site of pMT vector (see Materials and methods section). Available from Akihisa Terakita lab. |

*Continued on next page*

*Continued*

| Reagent type (species) or resource | Designation | Source or reference | Identifiers | Additional information |
|---|---|---|---|---|
| Recombinant DNA reagent | pUSRα-Antho2e_1D4 | This study | | The CDS of Antho2e was tagged with rho 1D4 epitope and inserted into the multicloning site of pUSRα vector (see Materials and methods section). Available from Akihisa Terakita lab. |
| Recombinant DNA reagent | pMT-Antho2c_1D4 | This study | | The CDS of Antho2e was tagged with rho 1D4 epitope and inserted into the multicloning site of pMT vector (see Materials and methods section). Available from Akihisa Terakita lab. |
| Recombinant DNA reagent | pcDNA3.1+/mit-2mutAEQ | Addgene | RRID:Addgene_45539 | |
| Cell line (African green monkey) | COS-1 | David Farrens lab. | RRID:CVCL_0223 | |
| Cell line (*Homo sapiens*) | Human embryonic kidney 293S (HEK293S) | | RRID:CVCL_A784 | |
| Commercial assay or kit | In-Fusion HD cloning kit | TAKARA | Cat no. 639650 | |
| Chemical compound, drug | PEI MAX - Transfection Grade Linear Polyethyleneimine Hydrochloride | Kyfora Bio | 24765 | |
| Chemical compound, drug | Dodecyl β-D-maltoside | DOJIMBO | D316-12 | |
| Software, algorithm | IGOR Pro 8 | https://www.wavemetrics.com/ | | |
| Software, algorithm | MAFFT v7 | *Katoh and Standley, 2013* | | |
| Software, algorithm | ModelTest-NG v0.2.0 | *Darriba et al., 2020* | | |
| Software, algorithm | RAxML-NG v1.2.0 | *Kozlov et al., 2019* | | |
| Software, algorithm | AlphaFold2 | *Jumper et al., 2021* | | |
| Software, algorithm | HomolWat | *Mayol et al., 2020* | | |
| Software, algorithm | PROPKA | *Olsson et al., 2011* | | |
| Software, algorithm | AMBER | *Case et al., 2025* | | |
| Software, algorithm | Orca 5.0.2 | *Neese, 2022* | | |
| Software, algorithm | ChemShell 3.7.1 | *Metz et al., 2014* | | |
| Software, algorithm | Turbomole 7.5.1 | *Furche et al., 2014* | | |
| Software, algorithm | PyMOL 2.5.5. | The PyMOL Molecular Graphics System, Version 2.5.5 Schrödinger, LLC | | |

## Experimental design

We first identified and cloned opsins from a reef-building coral, *A. tenuis*, and then expressed opsins belonging to the ASO-II group in mammalian cultured cells. We performed spectroscopic measurements of purified pigments of the opsins in different pH and Cl⁻ conditions to identify their effects on the acid dissociation constant of the protonated Schiff base of the opsins, leading to the determination of the counterion. Computational modeling and QM/MM calculations were also conducted to elucidate the retinylidene Schiff base environment in the dark state of Antho2a. Light-evoked $Ca^{2+}$ responses were assessed by aequorin-based bioluminescent reporter assay to evaluate the G protein activation of Antho2a.

## Identification of *A. tenuis* opsins and phylogenetic tree inference

*Acropora tenuis* (Dana, 1846) is a common reef-building coral distributed throughout the Indo-Pacific Ocean. Candidate sequences of *A. tenuis* opsins were identified by homology search against public genome and transcriptome datasets (*Shinzato et al., 2021*; *Voolstra et al., 2015*), and their phylogenetic relationships to known opsins were inferred by subsequent phylogenetic tree reconstruction. We first conducted BLASTP and TBLASTN searches with an E-value cutoff of $10^{-10}$ using *Acropora palmata* Acropsin 1–3 (JQ966100-JQ966102), two *Nematostella vectensis* opsins (BR000676-BR000677), human rhodopsin (NM_000539), and squid rhodopsin (X70498) as queries. We aligned the collected opsin homologs and excluded sequences that did not contain a retinal-binding lysine residue (Lys296) in the seventh transmembrane helix. We modified the fragmented sequences by reference to the genome sequence of *A. tenuis* and opsin sequences of other *Acropora* species (*A. palmata* or *A. millepora*). The candidate sequences of *A. tenuis* opsins were combined with the representative opsin sequences. The final sequence set was aligned using MAFFT (*Katoh and Standley, 2013*) and trimmed by TrimAl (*Capella-Gutiérrez et al., 2009*) with the 'gappyout' function. The ML tree was reconstructed using RAxML-NG v1.1.0 (*Kozlov et al., 2019*) assuming the LG+G4 model of protein evolution, which was selected by ModelTest-NG v0.2.0 (*Darriba et al., 2020*). The ML branch supports were estimated with 1000 bootstrap replicates.

## Sample collection, total RNA extraction, and cDNA synthesis

Colonies of *A. tenuis* were collected from <3 m depth on the fringing reef on Sesoko Island, Okinawa (N26°37.58′, E127°52.01′) and were maintained in flow-through aquaria at Sesoko Station (Tropical Biosphere Research Center, University of Ryukyus, Okinawa, Japan). Four days after spawning, motile larvae and small branches of adult colonies were preserved in RNAlater Stabilization Solution (Thermo Fisher Scientific, MA, USA). Total RNAs were extracted from the larval and adult samples using TRIzol reagent (Thermo Fisher Scientific) or Sepasol-RNA I Super G (nacalai tesque, Kyoto, Japan) and purified using QIAGEN RNeasy Mini Kit (QIAGEN, Hilden, Germany) following the manufacturer's protocol. cDNAs were synthesized from the total RNA by reverse transcription using High-Capacity cDNA Reverse Transcription kits (Thermo Fisher Scientific).

## Expression and purification of *A. tenuis* opsins

The coding regions of *A. tenuis* opsins were amplified by PCR with gene-specific primers and were tagged with the epitope sequence of the anti-bovine rhodopsin antibody rho 1D4 (ETSQVAPA) at their C-termini. Site-directed mutants were produced by overlap extension PCR using PrimeSTAR Max DNA Polymerase (TAKARA, Shiga, Japan) with site-specific primers and were also tagged with the 1D4 epitope sequence. The tagged cDNAs were inserted into the pUSRα vector (*Kayada et al., 1995*) digested with HindIII and EcoRI or the pMT vector (*Ridge and Abdulaev, 2000*) digested with EcoRI and NotI using In-Fusion HD cloning kit (TAKARA). The plasmids (15 µg per 100 mm culture dish) were transfected into COS-1 cells using the polyethyleneimine (PEI) transfection method as described previously (*Obayashi et al., 2025*; *Sinha et al., 2014*). The transfected cells were maintained for 24 hr after transfection at 37°C under 5% $CO_2$ and then 11-*cis* retinal was added to the medium (1 µL of 4 mM 11-*cis* retinal to 100 mm culture dish) following 25°C or 30°C incubation for another 24 hr in the dark before collecting the cells. The reconstituted pigments were extracted from the cell membranes with 1% dodecyl β-D-maltoside (DDM, Dojindo, Kumamoto, Japan), 50 mM HEPES, and 140 mM NaCl (pH 6.5). The solubilized samples were mixed with 1D4-conjugated agarose beads overnight, and the mixture was transferred into Bio-Spin columns (Bio-Rad, Hercules, CA, USA) and washed in the buffer containing 0.02% DDM, 50 mM HEPES, and 140 mM NaCl (pH 6.5, buffer A). The purified pigments were eluted with buffer A containing 0.5–1 mg/mL 1D4 peptide (custom peptide synthesis by GenScript Japan Inc, Tokyo, Japan). To obtain pigments in solutions of various anions ($SO_4^{2-}$, $Br^-$, and $I^-$) other than $Cl^-$, samples were prepared as described above and in the final step, the mixture of solubilized samples and 1D4-agarose beads was washed with buffer A followed by the additional wash with buffers including different sodium salts of anions (0.02% DDM, each of 70 mM $Na_2SO_4$, 140 mM NaBr, or 140 mM NaI, and 50 mM HEPES). Then, the pigments were eluted with the buffer including the appropriate sodium salt of anion containing 0.5–1 mg/mL 1D4 peptide. Alternatively, for some pigments that were unstable in the absence of $Cl^-$, we quickly removed $Cl^-$ by gel-filtration chromatography on PD MiniTrap desalting columns with Sephadex G-25 resin (Cytiva, Marlborough,

MA, USA). The columns were first equilibrated with the buffer including 0.02% DDM, 70 mM $Na_2SO_4$, and 50 mM HEPES, 500 μL of samples were loaded onto the columns and eluted with the buffer. We collected 800 μL fractions and used them for subsequent spectroscopic analyses.

## UV-visible spectroscopy

Spectroscopic measurements were performed at 0°C using a V-750 UV-visible spectrophotometer (JASCO Corporation, Tokyo, Japan). The pH of the samples was adjusted with 100 mM CAPS, including NaOH for alkaline conditions and 500 mM $NaH_2PO_4$ for acidic conditions. pH values were measured using a pH meter (B-211; HORIBA, Kyoto, Japan) immediately after each spectroscopic measurement. The concentration of $Cl^-$ in the samples was adjusted by addition of different concentrations of NaCl solutions which were prepared in 70 mM $Na_2SO_4$ buffer (see above). A 100 W halogen lamp was equipped on the spectrophotometer and used to illuminate samples with a set of optical interference filters (420 nm or 500 nm, Toshiba) and cutoff filters (O-55 or O-56, AGC Techno Glass Co., Shizuoka, Japan). Absorption spectra of some UV-absorbing pigments were recorded using the V-750 UV-visible spectrophotometer, equipped with a 300 W xenon lamp (MAX-350; Asahi Spectra Co., Tokyo, Japan) that was used for illumination of samples in combination with a UV-transmitting filter (UTVAF-50S-36U, SIGMA KOKI, Tokyo, Japan).

## HPLC analysis

An HPLC analysis was carried out to analyze the conformations of retinal present in the purified pigments as described previously (*Terakita et al., 1989*), with some modifications. Briefly, 100 μL of purified pigments were mixed with 210 μL of cold 90% methanol which was stored in −20°C and 30 μL of 1 M hydroxylamine to convert retinal chromophore in a sample into retinal oxime. The retinal oxime was extracted with 700 μL of *n*-hexane. 200 μL of the extract were injected into a YMC-Pack SIL column (particle size 3 μm, 150×6.0 mm²) and eluted with *n*-hexane containing 15% (vol/vol) ethyl acetate and 0.15% (vol/vol) ethanol at a flow rate of 1 mL/min.

## Bioluminescent reporter assays for $Ca^{2+}$ measurements in cultured cells

$Ca^{2+}$ levels in opsin-expressing cultured cells were assessed by an aequorin-based luminescent assay as described previously (*Koyanagi et al., 2022*). Briefly, the plasmid containing open reading frames of each opsin was transfected into HEK293S cells in 35 mm dishes by the PEI method with the aequorin plasmid obtained by introducing a reverse mutation A119D into the plasmid (pcDNA3.1+/mit-2mutAEQ) (Addgene no. 45539) (*de la Fuente et al., 2012*). The transfected HEK293S cells were incubated for ~24 hr at 37°C under 5% $CO_2$ with the addition of 0.2 μM/dish of 11-*cis* retinal 4–5 hr after the transfection. Before the luminescence measurements, the culture medium was replaced with a medium containing coelenterazine h (pH 7.0), and the cells were incubated to equilibrate with the media at 25°C for at least 2 hr. Dishes of cells were then stimulated with light, and luminescence values were recorded using GloMax 20/20n Luminometer (Promega). A green (495 nm) LED light (color: 'Cyan', Ex-DHC; BioTools Inc, Gunma, Japan) and arrayed LEDs on a board with spectral emission peaks at 390 nm and 510 nm (SPL-25-CC; REVOX Inc, Kanagawa, Japan) were used as light sources.

## Cell lines

The identities of the HEK293S cell line used in the study were authenticated by short-tandem repeat profiling. The COS-1 cells were kindly provided by Dr. David Farres (Oregon Health & Science University) and have been maintained in the laboratory. We have checked that both HEK293S cell and COS-1 cell lines were free from mycoplasma contamination using real-time PCR.

## Computational modeling and QM/MM calculations

The three-dimensional structure of Antho2a was predicted from the primary amino acid sequence using AlphaFold2 (*Jumper et al., 2021*). The 11-*cis* retinal chromophore linked to the protonated Schiff base was incorporated into the AlphaFold model using as a template the high-resolution structure of bovine rhodopsin solved by time-resolved serial femtosecond X-ray crystallography (*Gruhl et al., 2023*). A $Cl^-$ anion was initially placed in close proximity to the retinal protonated Schiff base, as observed in the microbial chloride-pump halorhodopsin (*Mous et al., 2022*). Water molecules were added using HomolWat (*Mayol et al., 2020*). We determined the p$K_a$ values of the titratable

amino acid residues at pH 6.5 using the PROPKA program (*Olsson et al., 2011*; *Søndergaard et al., 2011*) and subsequently, the protein was protonated using the tleap program in the AMBER software package (*Case et al., 2016*). The geometry of this initial model was first relaxed by molecular mechanics energy minimization with the Amber ff14SB force field (*Maier et al., 2015*) using steepest descent for 10,000 steps before switching to a conjugate gradient minimizer for an additional 10,000 steps. During energy minimization, a positional restraint of 10 kcal/mol/Å$^2$ was applied to all atoms, including hydrogens. The SHAKE algorithm (*Ryckaert et al., 1977*) was used to constrain the motion of bonds involving hydrogen. Finally, the geometry of the system was optimized using hybrid QM/MM calculations without considering any external environment and with the backbone of the protein frozen (*Mroginski et al., 2021*; *Senn and Thiel, 2009*). The QM part consists of the retinal chromophore linked to the lysine side chain cut between the Cδ and Cε atoms forming the protonated Schiff base, along with Cl$^-$, Glu292, and Ser186. The retinal-binding pocket also contains predicted water molecules (modeled based on homologous GPCR structures) close to the Schiff base and the chloride ion, which were not included in the QM region. The hydrogen link atom scheme was used at the QM/MM boundary. The QM part was treated using the BP86-D3 (BJ) functional (*Becke, 1988*; *Grimme et al., 2011*) in conjunction with the cc-pVDZ basis set (*Dunning, 1989*) and the def2/J auxiliary basis set for the resolution of identity (RI) (*Weigend, 2008*). The chain of spheres exchange algorithm was utilized in combination with the resolution of identity for the Coulomb term (RI-J). The rest of the protein was treated with the Amber ff14SB force field. Water molecules were treated with the TIP3P model (*Jorgensen et al., 1983*). The QM/MM calculations were performed using the quantum chemistry program Orca 5.0.2 (*Neese, 2022*) interfaced with the DL_POLY module of the ChemShell 3.7.1 software package (*Metz et al., 2014*; *Sherwood et al., 2003*). The optimized ground state geometries and partial charges were used to calculate the vertical excitation energies using the simplified time-dependent density functional theory (*Bannwarth and Grimme, 2014*; *Runge and Gross, 1984*) at the CAM-B3LYP/cc-pVTZ level of theory (*Dunning, 1989*; *Yanai et al., 2004*) using the Orca program. The excitation energies were also calculated using the RI-ADC(2) method (*Hättig, 2005*) with frozen core orbitals and the cc-pVTZ basis set in association with the corresponding auxiliary basis by utilizing the Turbomole 7.5.1 program package (*Furche et al., 2014*). The three-dimensional models were visualized using the molecular graphics program PyMOL 2.5.5.

## Acknowledgements

We thank Dr. Robert S Molday (University of British Columbia) for kindly supplying rho 1D4-producing hybridoma. We thank Dr. Masayuki Hatta (Ochanomizu University) for coral sampling. We thank Dr. David Farrens (Oregon Health & Science University) for kindly providing us with COS-1 cell line and are also grateful to Dr. Hisao Tsukamoto (Kobe University) for technical guidance on the maintenance and transfection of COS-1 cells. We finally thank the members of the High-Performance Computing and Emerging Technologies (HPCE) group at the Paul Scherrer Institute for technical support and assistance with high-performance computing. This work was supported by the Japanese Ministry of Education, Culture, Sports, Science and Technology Grants-in-Aid for Scientific Research 23H02516 (to AT), 22H02663 (to MK), and JP20J01841 (YS); and Japan Science and Technology Agency (JST) Core Research for Evolutional Science and Technology (CREST) Grant JPMJCR1753 (to AT). YS was supported by Grant-in-Aid for JSPS Fellows. This work has also been supported by the Swiss National Science Foundation (project grant 192780 to XD) and by the European Union's Horizon 2020 research and innovation programme (grant agreement 951644 to GFXS and Marie Skłodowska-Curie grant agreement 884104 [PSI-FELLOW-III-3i] to SS).

## Additional information

### Funding

| Funder | Grant reference number | Author |
| --- | --- | --- |
| Ministry of Education, Culture, Sports, Science and Technology | Grants-in-Aid for Scientific Research 23H02516 | Akihisa Terakita |

| Funder | Grant reference number | Author |
|---|---|---|
| Ministry of Education, Culture, Sports, Science and Technology | Grants-in-Aid for Scientific Research 22H02663 | Mitsumasa Koyanagi |
| Ministry of Education, Culture, Sports, Science and Technology | Grants-in-Aid for Scientific Research JP20J01841 | Yusuke Sakai |
| Japan Science and Technology Agency | Core Research for Evolutional Science and Technology (CREST) Grant JPMJCR1753 | Akihisa Terakita |
| Swiss National Science Foundation | project grant 192780 | Xavier Deupi |
| Horizon 2020 Framework Programme | 10.3030/951644 | Gebhard FX Schertler |
| Horizon 2020 Framework Programme | Marie Skłodowska-Curie grant agreement 884104 (PSI-FELLOW-III-3i) | Saumik Sen |

The funders had no role in study design, data collection and interpretation, or the decision to submit the work for publication.

### Author contributions

Yusuke Sakai, Conceptualization, Data curation, Formal analysis, Funding acquisition, Investigation, Methodology, Validation, Visualization, Writing – original draft, Writing – review and editing; Saumik Sen, Data curation, Formal analysis, Funding acquisition, Investigation, Methodology, Software, Writing – review and editing; Tomohiro Sugihara, Yukiya Kakeyama, Makoto Iwasaki, Investigation, Writing – review and editing; Gebhard FX Schertler, Funding acquisition, Supervision, Writing – review and editing; Xavier Deupi, Data curation, Formal analysis, Funding acquisition, Investigation, Methodology, Software, Supervision, Validation, Writing – review and editing; Mitsumasa Koyanagi, Conceptualization, Funding acquisition, Methodology, Supervision, Validation, Writing – review and editing; Akihisa Terakita, Conceptualization, Funding acquisition, Methodology, Project administration, Supervision, Validation, Writing – original draft, Writing – review and editing

### Author ORCIDs

Yusuke Sakai ⓘ https://orcid.org/0000-0002-7092-0094
Gebhard FX Schertler ⓘ https://orcid.org/0000-0002-5846-6810
Xavier Deupi ⓘ https://orcid.org/0000-0003-4572-9316
Mitsumasa Koyanagi ⓘ https://orcid.org/0000-0002-7741-918X
Akihisa Terakita ⓘ https://orcid.org/0000-0002-8379-8913

Reviewer #1 (Public review): https://doi.org/10.7554/eLife.105451.3.sa1
Reviewer #2 (Public review): https://doi.org/10.7554/eLife.105451.3.sa2
Reviewer #3 (Public review): https://doi.org/10.7554/eLife.105451.3.sa3
Author response https://doi.org/10.7554/eLife.105451.3.sa4

## Additional files

### Supplementary files
MDAR checklist

### Data availability
The cDNA sequences of Acropora tenuis opsins in this paper are available in GenBank: eight Cnidopsins (accession no. LC844924-LC844931), seven opsins in the ASO-II group (accession no. LC844932-LC844938), and one opsin in the ASO-I group (accession no. LC844939). The structural models of wild type Antho2a with a neutral or charged Glu292 and the Antho2a E292A mutant are

available in Zenodo (10.5281/zenodo.15064942). All data needed to evaluate the conclusion in this paper are present in the paper and the source data files.

The following datasets were generated:

| Author(s) | Year | Dataset title | Dataset URL | Database and Identifier |
|---|---|---|---|---|
| Deupi X, Sen S | 2025 | Theoretical structural models of the anthozoan-specific opsin II Antho2a (wild type with Glu292 either neutral or negatively charged and the E292A mutant). | https://doi.org/ 10.5281/zenodo. 15064943 | Zenodo, 10.5281/ zenodo.15064943 |
| Sakai Y, Sen S, Sugihara T, Kakeyama Y, Iwasaki M, Schertler GFX, Deupi X, Koyanagi M, Terakita A | 2025 | Acropora tenuis Cnidopsin1 CDS | https://www.ncbi.nlm. nih.gov/nuccore/? term=LC844924 | NCBI Nucleotide, LC844924 |
| Sakai Y, Sen S, Sugihara T, Kakeyama Y, Iwasaki M, Schertler GFX, Deupi X, Koyanagi M, Terakita A | 2025 | Acropora tenuis Cnidopsin2 CDS | https://www.ncbi.nlm. nih.gov/nuccore/? term=LC844925 | NCBI Nucleotide, LC844925 |
| Sakai Y, Sen S, Sugihara T, Kakeyama Y, Iwasaki M, Schertler GFX, Deupi X, Koyanagi M, Terakita A | 2025 | Acropora tenuis Cnidopsin3a CDS | https://www.ncbi.nlm. nih.gov/nuccore/? term=LC844926 | NCBI Nucleotide, LC844926 |
| Sakai Y, Sen S, Sugihara T, Kakeyama Y, Iwasaki M, Schertler GFX, Deupi X, Koyanagi M, Terakita A | 2025 | Acropora tenuis Cnidopsin3b CDS | https://www.ncbi.nlm. nih.gov/nuccore/? term=LC844927 | NCBI Nucleotide, LC844927 |
| Sakai Y, Sen S, Sugihara T, Kakeyama Y, Iwasaki M, Schertler GFX, Deupi X, Koyanagi M, Terakita A | 2025 | Acropora tenuis Cnidopsin4 CDS | https://www.ncbi.nlm. nih.gov/nuccore/? term=LC844928 | NCBI Nucleotide, LC844928 |
| Sakai Y, Sen S, Sugihara T, Kakeyama Y, Iwasaki M, Schertler GFX, Deupi X, Koyanagi M, Terakita A | 2025 | Acropora tenuis Cnidopsin5 CDS | https://www.ncbi.nlm. nih.gov/nuccore/? term=LC844929 | NCBI Nucleotide, LC844929 |
| Sakai Y, Sen S, Sugihara T, Kakeyama Y, Iwasaki M, Schertler GFX, Deupi X, Koyanagi M, Terakita A | 2025 | Acropora tenuis Cnidopsin6 CDS | https://www.ncbi.nlm. nih.gov/nuccore/? term=LC844930 | NCBI Nucleotide, LC844930 |
| Sakai Y, Sen S, Sugihara T, Kakeyama Y, Iwasaki M, Schertler GFX, Deupi X, Koyanagi M, Terakita A | 2025 | Acropora tenuis Cnidopsin7 CDS | https://www.ncbi.nlm. nih.gov/nuccore/? term=LC844931 | NCBI Nucleotide, LC844931 |

*Continued on next page*

*Continued*

| Author(s) | Year | Dataset title | Dataset URL | Database and Identifier |
|---|---|---|---|---|
| Sakai Y, Sen S, Sugihara T, Kakeyama Y, Iwasaki M, Schertler GFX, Deupi X, Koyanagi M, Terakita A | 2025 | Acropora tenuis Antho2a CDS | https://www.ncbi.nlm.nih.gov/nuccore/?term=LC844932 | NCBI Nucleotide, LC844932 |
| Sakai Y, Sen S, Sugihara T, Kakeyama Y, Iwasaki M, Schertler GFX, Deupi X, Koyanagi M, Terakita A | 2025 | Acropora tenuis Antho2b CDS | https://www.ncbi.nlm.nih.gov/nuccore/?term=LC844933 | NCBI Nucleotide, LC844933 |
| Sakai Y, Sen S, Sugihara T, Kakeyama Y, Iwasaki M, Schertler GFX, Deupi X, Koyanagi M, Terakita A | 2025 | Acropora tenuis Antho2c CDS | https://www.ncbi.nlm.nih.gov/nuccore/?term=LC844934 | NCBI Nucleotide, LC844934 |
| Sakai Y, Sen S, Sugihara T, Kakeyama Y, Iwasaki M, Schertler GFX, Deupi X, Koyanagi M, Terakita A | 2025 | Acropora tenuis Antho2d CDS | https://www.ncbi.nlm.nih.gov/nuccore/?term=LC844935 | NCBI Nucleotide, LC844935 |
| Sakai Y, Sen S, Sugihara T, Kakeyama Y, Iwasaki M, Schertler GFX, Deupi X, Koyanagi M, Terakita A | 2025 | Acropora tenuis Antho2e CDS | https://www.ncbi.nlm.nih.gov/nuccore/?term=LC844936 | NCBI Nucleotide, LC844936 |
| Sakai Y, Sen S, Sugihara T, Kakeyama Y, Iwasaki M, Schertler GFX, Deupi X, Koyanagi M, Terakita A | 2025 | Acropora tenuis Antho2f CDS | https://www.ncbi.nlm.nih.gov/nuccore/?term=LC844937 | NCBI Nucleotide, LC844937 |
| Sakai Y, Sen S, Sugihara T, Kakeyama Y, Iwasaki M, Schertler GFX, Deupi X, Koyanagi M, Terakita A | 2025 | Acropora tenuis Antho2g CDS | https://www.ncbi.nlm.nih.gov/nuccore/?term=LC844938 | NCBI Nucleotide, LC844938 |
| Sakai Y, Sen S, Sugihara T, Kakeyama Y, Iwasaki M, Schertler GFX, Deupi X, Koyanagi M, Terakita A | 2025 | Acropora tenuis Antho1 CDS | https://www.ncbi.nlm.nih.gov/nuccore/?term=LC844939 | NCBI Nucleotide, LC844939 |

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
