## [Editor Report · eLife Assessment]

The authors provide **compelling** evidence that a chloride ion stabilizes the protonated Schiff base chromophore linkage in the animal rhodopsin Antho2a. This **important** finding is novel and of major interest to a broad audience, including optogenetics researchers, protein engineers, spectroscopists, and environmental biologists. The study combines state-of-the-art research methods, such as spectroscopic and mutational analyses, which are complemented by QM/MM calculations, and was further improved based on the comments from the reviewers.

---

## [Referee Report · Reviewer #1 (Public review)]

The chromophore molecule of animal and microbial rhodopsins is retinal which forms a Schiff base linkage with a lysine in the 7-th transmembrane helix. In most cases, the chromophore is positively charged by protonation of the Schiff base, which is stabilized by a negatively charged counterion. In animal opsins, three sites have been experimentally identified, Glu94 in helix 2, Glu113 in helix 3, and Glu181 in extracellular loop 2, where a glutamate acts as the counterion by deprotonation. In this paper, Sakai et al. investigated molecular properties of anthozoan-specific opsin II (ASO-II opsins), as they lack these glutamates. They found an alternative candidate, Glu292 in helix 7, from the sequences. Interestingly, the experimental data suggested that Glu292 is not the direct counterion in ASO-II opsins. Instead, they found that ASO-II opsins employ a chloride ion as the counterion. In case of microbial rhodopsin, a chloride ion serves as the counterion of light-driven chloride pumps. This paper reports the first observation of a chloride ion as the counterion in animal rhodopsin. Theoretical calculation using a QM/MM method supports their experimental data. The authors also revealed the role of Glu292, which serves as the counterion in the photoproduct and is involved in G protein activation.

The conclusions of this paper are well supported by data.

---

## [Referee Report · Reviewer #2 (Public review)]

Summary:

This work reports the discovery of a new rhodopsin from reef-building corals that is characterized experimentally, spectroscopically, and by simulation. This rhodopsin lacks a carboxylate-based counterion, which is typical for this family of proteins. Instead, the authors find that a chloride ion stabilizes the protonated Schiff base and thus serves as a counterion.

Strengths:

This work focuses on the rhodopsin Antho2a, which absorbs in the visible spectrum with a maximum at 503 nm. Spectroscopic studies under different pH conditions, including the mutant E292A and different chloride concentrations, indicate that chloride acts as a counterion in the dark. In the photoproduct, however, the counterion is identified as E292.

These results lead to a computational model of Antho2a in which the chloride is modeled in addition to the Schiff base. This model is improved using the hybrid QM/MM simulations. As a validation, the absorption maximum is calculated using the QM/MM approach for the protonated and deprotonated E292 residue as well as the E292A mutant. The results are in good agreement with the experiment. However, there is a larger deviation for ADC(2) than for sTD-DFT. Nevertheless, the trend is robust since the wt and E292A mutant models have similar excitation energies. The calculations are performed at a high level of theory that includes a large QM region.

---

## [Referee Report · Reviewer #3 (Public review)]

Summary:

The paper by Saito et al. studies the properties of anthozoan-specific opsins (ASO-II) from organisms found in reef-building coral. Their goal was to test if ASO-II opsins can absorb visible light, and if so, what are they key factors involved.

The most exciting aspect of this work is their discovery that ASO-II opsins do not have a counterion residue (Asp or Glu) located at any of the previously known sites found in other animal opsins.

This is very surprising. Opsins are only able to absorb visible (long wavelength light) if the retinal Schiff base is protonated, and the latter requires (as the name implies) a "counter ion". However, the authors clearly show that some ASO-II opsins do absorb visible light.

To address this conundrum, they tested if the counterion could be provided by exogenous chloride ions (Cl-). Their results find compelling evidence supporting this idea, and their studies of ASO-II mutant E292A suggests E292 also plays a role in G protein activation and is a counterion for a protonated Schiff base in the light-activated form.

Strengths:

Overall, the methods are well described and carefully executed, and the results very compelling.

Their analysis of seven ASO-II opsin sequences undoubtedly shows they all lack a Glu or Asp residue at "normal" (previously established) counter-ion sites in mammalian opsins (typically found at positions 94, 113 or 181). The experimental studies clearly demonstrate the necessity of Cl- for visible light absorbance, as do their studies of the effect of altering the pH.

Importantly, the authors also carried out careful QM/MM computational analysis (and corresponding calculation of the expected absorbance effects), thus providing compelling support for the Cl- acting directly as a counterion to the protonated retinal Schiff base, and thus limiting the possibility that the Cl- is simply altering the absorbance of ASO-II opsins through some indirect effect on the protein.

Altogether, the authors clearly achieved their aims, and the results support their conclusions. The manuscript is carefully written, and refreshingly, the results and conclusions not overstated.

This study is impactful for several reasons. There is increasing interest in optogenetic tools, especially those that leverage G protein coupled receptor systems. Thus, the authors demonstration that ASO-II opsins could be useful for such studies is of interest.

Moreover, the finding that visible light absorbance by an opsin does not absolutely require a negatively charged amino acid be placed at one of the expected sites (94, 113 or 181) typically found in animal opsins is very intriguing and will help future protein engineering efforts. The argument that the Cl- counterion system they discover here might have been a preliminary step in the evolution of amino acid based counterions used in animal opsins is also interesting.

Finally, given the ongoing degradation of coral reefs worldwide, the focus on these curious opsins is very timely, as is the authors proposal that the lower Schiff base pKa they discovered here for ASO-II opsins may cause them to change their spectral sensitivity and G protein activation due to changes in their environmental pH.

---

## [Author Response]

The following is the authors’ response to the original reviews

**Public Reviews:**

**Reviewer #1 (Public review):**
The chromophore molecule of animal and microbial rhodopsins is retinal which forms a Schiff base linkage with a lysine in the 7-th transmembrane helix. In most cases, the chromophore is positively charged by protonation of the Schiff base, which is stabilized by a negatively charged counterion. In animal opsins, three sites have been experimentally identified, Glu94 in helix 2, Glu113 in helix 3, and Glu181 in extracellular loop 2, where a glutamate acts as the counterion by deprotonation. In this paper, Sakai et al. investigated molecular properties of anthozoan-specific opsin II (ASO-II opsins), as they lack these glutamates. They found an alternative candidate, Glu292 in helix 7, from the sequences. Interestingly, the experimental data suggested that Glu292 is not the direct counterion in ASO-II opsins. Instead, they found that ASO-II opsins employ a chloride ion as the counterion. In the case of microbial rhodopsin, a chloride ion serves as the counterion of light-driven chloride pumps. This paper reports the first observation of a chloride ion as the counterion in animal rhodopsin. Theoretical calculation using a QM/MM method supports their experimental data. The authors also revealed the role of Glu292, which serves as the counterion in the photoproduct, and is involved in G protein activation.The conclusions of this paper are well supported by data, while the following aspects should be considered for the improvement of the manuscript.

We thank the reviewer for carefully reading the manuscript and providing important suggestions. Below, we address the specific comments.

(1) Information on sequence alignment only appears in Figure S2, not in the main figures. Figure S2 is too complicated by so many opsins and residue positions. It will be difficult for general readers to follow the manuscript because of such an organization. I recommend the authors show key residues in Figure 1 by picking up from Figure S2.

We thank the reviewer for pointing this out. As suggested, we have selected key residues (potential counterion sites) from Fig. S2 and show them now as Fig. 1B in the revised manuscript. Fig. S2 has also been simplified by showing only the most important residues.

(2) Halide size dependence. The authors observed spectral red-shift for larger halides. Their observation is fully coincident with the chromophore molecule in solution (Blatz et al. Biochemistry 1972), though the isomeric states are different (11-cis vs all-trans). This suggests that a halide ion is the hydrogen-bonding acceptor of the Schiff base N-H group in solution and ASO-II opsins. A halide ion is not the hydrogen-bonding acceptor in the structure of halorhodopsin, whose halide size dependence is not clearly correlated with absorption maxima (Scharf and Engelhard, Biochemistry 1994). These results support their model structure (Figure 4), and help QM/MM calculations.

We appreciate the comment, which provides a deeper insight into our results and reinforces our conclusions. We have revised the discussion of the effect of halide size on the λ_max_ shift to cite the prior work mentioned by the reviewer.

(3) QM/MM calculations. According to Materials and Methods, the authors added water molecules to the structure and performed their calculations. However, Figure 4 does not include such water molecules, and no information was given in the manuscript. In addition, no information was given for the chloride binding site (contact residues) in Figure 4. More detailed information should be shown with additional figures in Figure SX.

We thank the reviewer for making us realize that Fig. 4 was oversimplified.

We have added following text in the “Structural modelling and QM/MM calculations of the dark state of Antho2a” section:

Lines 220 – 223

“The chloride ion is also coordinated by two water molecules and the backbone of Cys187 which is part of a conserved disulfide bridge (Fig. S2). The retinylidene Schiff base region also includes polar (Ser186, Tyr91) and non-polar (Ala94, Leu113) residues (Fig. 4).”

We have updated Fig. 4 and its legend to show a more detailed environment of the protonated Schiff base and the chloride ion, including water molecules and other nearby residues.

(4) Figure 5 clearly shows much lower activity of E292A than that of WT, whose expression levels are unclear. How did the authors normalize (or not normalize) expression levels in this experiment?

We thank the reviewer for this valuable comment. In the previous version of the manuscript, we did not normalize the activity based on expression levels. We have considered this in the amended version.

First, we evaluated the expression levels of wild type and E292A Antho2a by comparing absorbances at λ_max_ (± 5 nm) of these pigments that were expressed and purified under the same conditions. Assuming that their molar absorption coefficients at the absorption maximum wavelengths are approximately the same, this can allow us to roughly compare their expression levels. The relative expression of the E292A mutant compared to the wild type (set as 1) was 0.81 at pH 6.5 and 140 mM NaCl, in which 94.0% (for E292A) and 99.8% (for wild type) of the Schiff base is protonated (Fig. 3A and B). As we conducted the live cell Ca^2+^ assay in media at pH 7.0, we estimated the proportion of the protonated states of wild type and E292A mutant at same pH. The relative amounts of the protonated states to the wild type at pH 6.5 (set as 1) were estimated to be 0.99 for wild type and 0.84 for E292A. Together, the protonated pigment of the E292A mutant was calculated to be about 73% of that of the wild type at pH 7.0. From Fig. 5, the amplitude of Ca^2+^ response of the E292A mutant was 12.1% of the wild type, showing that even after normalizing the expression levels, the Ca^2+^ response amplitude was lower in the E292A mutant than in the wild type. This leads to our conclusion that the E292A mutation can also influence the G protein activation efficiency.

We have added Fig. S11 showing the comparison of expression levels between the wild type and E292A of Antho2a (Fig. S11A) and maximum Ca^2+^ responses after normalizing the expression levels (Fig. S11B).

We have also revised the discussion section as follows:

Lines 324 – 335

“The relative expression level of the E292A mutant of Antho2a was approximately 0.81 of the wild type (set as 1), as determined by comparing absorbances at λ_max_ for both pigments expressed and purified under identical conditions (Fig. S11A). Additionally, the fraction of protonated pigment relative to the wild type (set as 1 at pH 6.5) was estimated to be 0.94 for the E292A mutant at pH 6.5, and 0.99 and 0.84 for the wild type and the E292A mutant at pH 7.0, respectively (Fig. 3A and B). Since pH 7.0 corresponds to the conditions used in the live cell Ca^2+^ assays, the effective amount of protonated pigment for the E292A mutant was approximately 73% of the wild type. Nevertheless, even after normalization for these differences, the Ca^2+^ response amplitude of the E292A mutant remained significantly lower (~ 17% of wild type, compared to the observed 12% prior to normalization; Fig. 5 and Fig. S11B). These observations suggest that Glu292 serves not only as a counterion in the photoproduct but also plays an allosteric role in influencing G protein activation.”

(5) The authors propose the counterion switching from a chloride ion to E292 upon light activation. A schematic drawing on the chromophore, a chloride ion, and E292 (and possible surroundings) in Antho2a and the photoproduct will aid readers' understanding.

We thank the reviewer for this excellent suggestion. We have prepared a new figure with a schematic drawing of the environment of the protonated Schiff base depicting the counterion switch in Fig. S10.

**Reviewer #2 (Public review):**
Summary:This work reports the discovery of a new rhodopsin from reef-building corals that is characterized experimentally, spectroscopically, and by simulation. This rhodopsin lacks a carboxylate-based counterion, which is typical for this family of proteins. Instead, the authors find that a chloride ion stabilizes the protonated Schiff base and thus serves as a counterion.Strengths:This work focuses on the rhodopsin Antho2a, which absorbs in the visible spectrum with a maximum at 503 nm. Spectroscopic studies under different pH conditions, including the mutant E292A and different chloride concentrations, indicate that chloride acts as a counterion in the dark. In the photoproduct, however, the counterion is identified as E292.These results lead to a computational model of Antho2a in which the chloride is modeled in addition to the Schiff base. This model is improved using the hybrid QM/MM simulations. As a validation, the absorption maximum is calculated using the QM/MM approach for the protonated and deprotonated E292 residue as well as the E292A mutant. The results are in good agreement with the experiment. However, there is a larger deviation for ADC(2) than for sTD-DFT. Nevertheless, the trend is robust since the wt and E292A mutant models have similar excitation energies. The calculations are performed at a high level of theory that includes a large QM region.Weaknesses:I have a couple of questions about this study:

We thank the reviewer for providing critical comments, particularly on the QM/MM calculations. We have carefully considered all comments and have addressed them as detailed below. Corresponding revisions have been made to the manuscript.

(1) I find it suspicious that the absorption maximum is so close to that of rhodopsin when the counterion is very different. Is it possible that the chloride creates an environment for the deprotonated E292, which is the actual counterion?

We think it is unlikely that the chloride ion merely facilitates deprotonation of Glu292 in such a way that it acts as the counterion of the dark state Antho2a. This conclusion is based on two results from our study. (1) λ_max_ of wild type Antho2a in the dark is positively correlated with the ionic radius of the halide in the solution; the λ_max_ is red shifted in the order Cl- < Br- < I- (Fig. 2E and F in the revised manuscript). This tendency is observed when the halide anion acts as a counterion of the protonated Schiff base (Blatz et al. Biochemistry 11: 848–855, 1972). (2) The QM/MM models of the dark state of Antho2a show that the calculated λ_max_ of Antho2a with a protonated (neutral) Glu292 is much closer to the experimentally observed λ_max_ than with a deprotonated (negatively charged) Glu292 (Fig. 4), suggesting that the Glu292 is likely to be protonated even in the presence of chloride ion. Therefore, we conclude that a solute anion, and not Glu292, acts as the counterion of the protonated Schiff base in the dark state of Antho2a. We have discussed this in the revised manuscript as follows:

Lines 274 – 291

“We found that the type of halide anions in the solution has a small but noticeable effect on the λ_max_ values of the dark state of Antho2a. This is consistent with the effect observed in a counterion-less mutant of bovine rhodopsin, in which halide ions serve as surrogate counterions (Nathans, 1990; Sakmar et al., 1991). Similarly, our results align with earlier observations that the λ_max_ of a retinylidene Schiff base in solution increases with the ionic radius of halides acting as hydrogen bond acceptors (i.e., I− > Br− > Cl−) (Blatz et al., 1972). In contrast, the λ_max_ of halorhodopsin from Natronobacterium pharaonic does not clearly correlate with halide ionic radius (Scharf and Engelhard, 1994), as the halide ion in this case is not a hydrogen-bonding acceptor of the protonated Schiff base (Kouyama et al., 2010; Mizuno et al., 2018). Altogether, these findings support our hypothesis that in Antho2a, a solute halide ion forms a hydrogen bond with the Schiff base, thereby serving as the counterion in the dark state. Moreover, QM/MM calculations for the dark state of Antho2a suggest that Glu292 is protonated and neutral, further supporting the hypothesis that Glu292 does not serve as the counterion in the dark state. However, unlike dark state, Cl− has little to no effect on the visible light absorption of the photoproduct (Fig. S5). Therefore, we conclude that Cl− and Glu292, respectively, act as counterions for the protonated Schiff base of the dark state and photoproduct of Antho2a. This represents a unique example of counterion switching from exogeneous anion to a specific amino acid residue upon light irradiation (Fig. S10).”

(2) The computational protocol states that water molecules have been added to the predicted protein structure. Are there water molecules next to the Schiff base, E292, and Cl-? If so, where are they located in the QM region?

We have updated Fig. 4 to show amino acids and water molecules near the Schiff base, E292, and the chloride ion. These include Ser186, Tyr91, Ala94, Leu113, Cys187, and two water molecules coordinating the chloride ion. We have added following text in the “Structural modelling and QM/MM calculations of the dark state of Antho2a” section of the revised manuscript.

Lines 220 – 223

“The chloride ion is also coordinated by two water molecules and the backbone of Cys187 which is part of a conserved disulfide bridge (Fig. S2). The retinylidene Schiff base region also includes polar (Ser186, Tyr91) and non-polar (Ala94, Leu113) residues (Fig. 4).”

Water molecules, which have been modelled by homology to other GPCR structures, were not included in the QM region. In the revised version of the manuscript, we clarify this point in the “Computational modelling and QM/MM calculations” section as follows.

Lines 515 – 517

“The retinal-binding pocket also contains predicted water molecules (modelled based on homologous GPCR structures) close to the Schiff base and the chloride ion which were not included in the QM region.”

(3) If the E292 residue is the counterion in the photoproduct state, I would expect the retinal Schiff base to rotate toward this side chain upon isomerization. Can this be modeled based on the recent XFEL results on rhodopsin?

The recent XFEL studies of rhodopsin reveal that at very early stages (1 ps after photoactivation), structural changes in retinal are limited primarily to the isomerization around the C11=C12 bond of the polyene chain, without significant rotation of the Schiff base.

Although modelling of a later active state with planar retinal and a rotated Schiff base is feasible—e.g., guided by high-resolution structures of bovine rhodopsin’s Meta II state such as PDB ID: 3PQR, see Author response image 1 below—active states of GPCRs typically exhibit substantial conformational flexibility and heterogeneity, making the generation of precise structural models suitable for accurate QM/MM calculations challenging. Despite these uncertainties, this preliminary modelling does indicate that upon isomerization to the all-trans configuration, the retinal Schiff base would rotate towards E292, supporting our hypothesis that E292 serves as the counterion in the Antho2a photoproduct. This is now shown better in the revised Fig. S10.

**Author response image 1. sa4fig1:** 

**Reviewer #3 (Public review):**
Summary:The paper by Saito et al. studies the properties of anthozoan-specific opsins (ASO-II) from organisms found in reef-building coral. Their goal was to test if ASO-II opsins can absorb visible light, and if so, what the key factors involved are.The most exciting aspect of this work is their discovery that ASO-II opsins do not have a counterion residue (Asp or Glu) located at any of the previously known sites found in other animal opsins.This is very surprising. Opsins are only able to absorb visible (long wavelength light) if the retinal Schiff base is protonated, and the latter requires (as the name implies) a "counter ion". However, the authors clearly show that some ASO-II opsins do absorb visible light.To address this conundrum, they tested if the counterion could be provided by exogenous chloride ions (Cl-). Their results find compelling evidence supporting this idea, and their studies of ASO-II mutant E292A suggest E292 also plays a role in G protein activation and is a counterion for a protonated Schiff base in the light-activated form.Strengths:Overall, the methods are well-described and carefully executed, and the results are very compelling.Their analysis of seven ASO-II opsin sequences undoubtedly shows they all lack a Glu or Asp residue at "normal" (previously established) counter-ion sites in mammalian opsins (typically found at positions 94, 113, or 181). The experimental studies clearly demonstrate the necessity of Cl- for visible light absorbance, as do their studies of the effect of altering the pH.Importantly, the authors also carried out careful QM/MM computational analysis (and corresponding calculation of the expected absorbance effects), thus providing compelling support for the Cl- acting directly as a counterion to the protonated retinal Schiff base, and thus limiting the possibility that the Cl- is simply altering the absorbance of ASO-II opsins through some indirect effect on the protein.Altogether, the authors achieved their aims, and the results support their conclusions. The manuscript is carefully written, and refreshingly, the results and conclusions are not overstated.This study is impactful for several reasons. There is increasing interest in optogenetic tools, especially those that leverage G protein-coupled receptor systems. Thus, the authors' demonstration that ASO-II opsins could be useful for such studies is of interest.Moreover, the finding that visible light absorbance by an opsin does not absolutely require a negatively charged amino acid to be placed at one of the expected sites (94, 113, or 181) typically found in animal opsins is very intriguing and will help future protein engineering efforts. The argument that the Cl- counterion system they discover here might have been a preliminary step in the evolution of amino acid based counterions used in animal opsins is also interesting.Finally, given the ongoing degradation of coral reefs worldwide, the focus on these curious opsins is very timely, as is the authors' proposal that the lower Schiff base pKa they discovered here for ASO-II opsins may cause them to change their spectral sensitivity and G protein activation due to changes in their environmental pH.

We thank the reviewer for the comprehensive summary of the manuscript and for finding it well-described and impactful.

**Recommendations for the Authors:**

**Reviewer #1 (Recommendations for the authors):**
(1) p. 5, l. 102: The authors obtained three absorption spectra out of seven. Did the authors examine the reasons for no absorption spectra for the remaining four proteins?

We have not identified the reasons for the absence of detectable absorption spectra for the remaining four opsins. We speculate that this could result from poor retinal binding under detergent-solubilized conditions, but we have not directly tested this possibility.

(2) p. 7, l. 141: The pH value is 7.5 in the text and 7.4 in Figure S4B.

We thank the reviewer for finding this mistake. The correct value is 7.4 and we have revised the text accordingly.

**Reviewer #2 (Recommendations for the authors):**
The structures and the simulations should be made available to the reader by providing them in a repository.

We have deposited the Antho2a models in Zenodo (https://zenodo.org/; an open-access repository for research data). We have added the following description in the “Data and materials availability” section of the revised manuscript.

Lines 559 – 560

“The structural models of wild type Antho2a with a neutral or charged Glu292 and the Antho2a E292A mutant are available in Zenodo (10.5281/zenodo.15064942).”

**Reviewer #3 (Recommendations for the authors):**
(1) In the homology models for the ASO-II opsins, are there any other possible residues that could act as counter-ion residues outside of the "normal" positions at 94, 113, or 181?

We have updated Fig. 4 to show all residues near the retinylidene Schiff base region, which include Cl−, Glu292, Ser186, Tyr91, Ala94, Leu113, Cys187, and two water molecules.

Apart from Cl− and Glu292, the homology models of the ASO-II opsins do not reveal any other candidate as the counterion of Schiff base. This is also suggested by the sequence alignment between opsins of the ASO-II group and other animal opsins in Fig. S2, where we show amino acid residues near the Schiff base (in addition to key motifs important for G protein activation).

(2) It is mentioned that the ASO-II opsins do not appear to be bistable opsins in detergents - do these opsins show any ability to photo-switch back and forth when in cellular membranes?

We have not directly tested whether Antho2a exhibits photo-switching in cellular membranes due to technical limitations associated with high light scattering in spectroscopic measurements. Instead, we recorded absorption spectra from crude extracts of detergent-solubilized cell membranes expressing Antho2a wild type (without purification) in the dark and after sequential light irradiation (Fig. S3C). This approach, which retains cellular lipids, can better preserve the photochemical properties of opsins, such as thermal stability and photoreactivity of their photoproducts, similar to intact cellular membranes. The first irradiation with green light (500 nm) led to a decrease in absorbance around the 550 nm region and an increase around the 450 nm region, indicating the formation of a photoproduct, consistent with observations using purified Antho2a.

However, subsequent irradiation with violet light (420 nm) did not reverse these spectral changes but resulted in only a slight decrease in absorbance around 400 nm. Re-exposure to green light produced no further spectral changes aside from baseline distortions. These findings suggest that the Antho2a photoproduct has limited ability to revert to its original dark state under these conditions. Nevertheless, because detergent solubilization may influence these observations, further studies in intact cellular membranes using live-cell assay will be required to conclusively assess bistability or photo-switching properties.

(3) The idea that E292 acts as a counterion for the protonated active state is intriguing - do the authors think the retinal decay process after light activation occurs with hydrolysis of the non-protonated form with subsequent retinal release?

We thank the reviewer for raising this important question. We first examined whether the increased UV absorbance observed after incubating the photoproduct for 20 hours in the dark (Fig. S3D, E, violet curves) originated from free retinal released from the opsin pigment. Acid denaturation (performed at pH 1.9) of this photoproduct resulted in a main product absorbing around 400 nm (Fig. S3G). Typically, when retinal binds opsin via the Schiff base (whether protonated or deprotonated), acid denaturation traps the retinal chromophore as a protonated Schiff base, yielding an absorption spectrum with a λ_max_ at approximately 440 nm, as observed in the dark state of Antho2a (Fig. S3F). Our results thus indicate that the UV absorbance in the photoproduct did not result from a deprotonated Schiff base but rather from retinal released during incubation. We have not directly tested whether the protonated or deprotonated form is more prone to retinal release. However, the decay of visible absorbance (associated with the protonated photoproduct) occurred more rapidly under alkaline conditions (pH 8.0), which generally favors deprotonation of the Schiff base (Fig. S3H). Thus, it is possible that the deprotonated photoproduct releases retinal more rapidly than the protonated form, but further studies are necessary to confirm this hypothesis.

To answer the comments (2) and (3) by the reviewer, we have added new panels (C and F–H) to Fig. S3.

We have revised the Results section as follows:

Lines 136 – 141

“The photoproduct remained stable for at least 5 minutes (Fig. S3A, curves 2 and 3) but did not revert to the original dark state upon subsequent irradiation (Fig. S3A and C). Instead, it underwent gradual decay accompanied by retinal release over time (Fig. S3D–G). These findings indicate that purified Antho2a is neither strictly bleach resistant nor bistable (see also Fig. S3 legend). We also observed that the protonated photoproduct decayed more rapidly at pH 8.0 (Fig. S3H) than at pH 6.5 (Fig. 3A, D, E).”

Text:(4) Page 3, line 38. Consider defining eumetazoan (for lay readers).

As suggested, we have defined eumetazoans and revised the sentence as follows:

Lines 38 – 40

“Opsins are present in the genomes of all eumetazoans (i.e., all animal lineages except sponges), and based on their phylogenetic relationships, they can be classified into eight groups…”

(5) Page 3, line 42. "But, furthermore, ..." should be changed to either word alone.

Revised as suggested.

(6) Page 18, line 447. The HPLC method is well-described and helpful. If possible, please add a Reference, or indicate if this is a new variation of the method.

This is a well-established method for analyzing the composition of retinal isomers bound to different states of rhodopsin pigments. We have now cited a reference describing the methodology (Terakita et al. Vision Res. 6: 639–652, 1989).

(7) Page 11, line 267. "..type of halide anions in the solution affected the λ_max_ values of the dark state of".Since the changes are not large (but clearly occur), consider changing this sentence to "..type of halide anions in the solution has a small but visible effect on the λ_max_ values of the dark state ..."

We have revised this sentence as suggested.

Figures:(9) Consider combining Figure FS6 with Figure 2 (effect of anions on visible absorbance).

As suggested, the previous Fig. S6 has been included in the main text as Fig. 2E and F in the revised manuscript.